# APP modulates KCC2 expression and function in hippocampal GABAergic inhibition

**Ming Chen[1], Jinzhao Wang[1], Jinxiang Jiang[2,3], Xingzhi Zheng[1], Nicholas J Justice[4], Kun Wang[1], Xiangqian Ran[1], Yi Li[1], Qingwei Huo[1], Jiajia Zhang[1], Hongmei Li[5], Nannan Lu[1], Ying Wang[1], Hui Zheng[5], Cheng Long[1,3], Li Yang[2,3,6]\***

[1]School of Life Sciences, South China Normal University, Guangzhou, China; [2]School of Psychology and Center for Studies of Psychological Application, South China Normal University, Guangzhou, China; [3]Brain Science Institute, South China Normal University, Guangzhou, China; [4]Institute of Molecular Medicine, University of Texas Health Sciences Center, Houston, United States; [5]Huffington Center on Aging, Baylor College of Medicine, Houston, United States; [6]Guangdong Key Laboratory of Mental Health and Cognitive Science, South China Normal University, Guangzhou, China

**\*For correspondence:** yang_li@m.
scnu.edu.cn

**Competing interests:** The
authors declare that no
competing interests exist.

**Reviewing editor:** Eunjoon Kim,
Institute for Basic Science, Korea
Advanced Institute of Science
and Technology, Republic of
Korea

**Abstract** Amyloid precursor protein (APP) is enriched at the synapse, but its synaptic function is still poorly understood. We previously showed that GABAergic short-term plasticity is impaired in *App* knock-out (*App*[-/-]) animals, but the precise mechanism by which APP regulates GABAergic synaptic transmission has remained elusive. Using electrophysiological, biochemical, moleculobiological, and pharmacological analysis, here we show that APP can physically interact with KCC2, a neuron-specific $K^+$-$Cl^-$ cotransporter that is essential for $Cl^-$ homeostasis and fast GABAergic inhibition. APP deficiency results in significant reductions in both total and membrane KCC2 levels, leading to a depolarizing shift in the GABA reversal potential ($E_{GABA}$). Simultaneous measurement of presynaptic action potentials and inhibitory postsynaptic currents (IPSCs) in hippocampal neurons reveals impaired unitary IPSC amplitudes attributable to a reduction in $\alpha 1$ subunit levels of $GABA_AR$. Importantly, restoration of normal KCC2 expression and function in *App*[-/-] mice rescues $E_{GABA}$, $GABA_AR$ $\alpha 1$ levels and $GABA_AR$ mediated phasic inhibition. We show that APP functions to limit tyrosine-phosphorylation and ubiquitination and thus subsequent degradation of KCC2, providing a mechanism by which APP influences KCC2 abundance. Together, these experiments elucidate a novel molecular pathway in which APP regulates, via protein-protein interaction with KCC2, $GABA_AR$ mediated inhibition in the hippocampus.

## Introduction

APP is a type I single pass transmembrane protein highly expressed in the central nervous system (CNS) which is processed by α-, β-, and γ-secretases (*Hardy and Selkoe, 2002*) to generate beta amyloid (Aβ) peptide fragments that comprise the amyloid plaques found in Alzheimer's disease (AD) patients post-mortem. The deleterious impact of Aβ has been studied extensively, with the recent discovery of the potent toxicity of Aβ oligomers on neuronal and synaptic activities (*Hsieh et al., 2006*; *Kim et al., 2006*; *Wei et al., 2010*). Despite advances in our understanding of how Aβ generation drives AD progression, the physiologic role of APP has proven more difficult to elucidate (*De Strooper and Annaert, 2000*). APP has been detected in vesicular fractions of

**eLife digest** Alzheimer's disease is the most common form of dementia. One of the hallmarks of the disease is the formation of sticky protein clumps called amyloid plaques in the brain. These plaques are formed from specific fragments of a protein called APP. The intact form of APP is essential for synapses (the junctions across which neurons transmit signals) to form and work correctly.

The hippocampus is one of the first brain regions to be affected in Alzheimer's disease and is important for forming memories and emotions. In the hippocampus, GABA$_A$ receptors at synapses normally tightly regulate synaptic signaling by reducing the ability of the receiving neuron to respond, but this inhibition is disrupted in Alzheimer's disease. Studies suggest that APP can affect how GABA$_A$ receptors transmit signals, but it is not known how it does so. One possibility is that APP regulates a protein called KCC2 that helps to maintain the inhibitory effect of GABA$_A$ receptors.

To investigate this, Chen et al. genetically modified mice to lack the gene that produces APP. These mice had a lower level of KCC2 in their hippocampus than normal mice, and their GABA$_A$ receptors were less able to inhibit synaptic signaling. Further experiments demonstrated that APP physically interacts with KCC2 and maintains normal levels of the protein by preventing it from being chemically modified and degraded.

Chen et al. also showed that treating mice that lack APP with specific compounds can restore KCC2 activity and return the behavior of synaptic GABA$_A$ receptors to normal. Future studies in mice (and eventually people) that exhibit symptoms of Alzheimer's disease will help to determine whether KCC2 is important in the development of the disease. If so, modifying the levels of the KCC2 protein in the brain could potentially help to slow down memory loss in Alzheimer's disease.

dendrites and axons (*Schubert et al., 1991*), suggesting a role for APP in synaptic activity. Moreover, we have shown that APP is required for normal adult neurogenesis (*Wang et al., 2014a*) and formation of the neuromuscular junction (*Wang et al., 2009*; *Yang et al., 2007*).

Many lines of evidence have suggested that APP function can impact the electrophysiological properties of neurons (*Kamenetz et al., 2003*; *Klevanski et al., 2015*; *Palop and Mucke, 2010*; *Priller et al., 2006*; *Selkoe, 2002*; *Wang et al., 2014a*; *Yang et al., 2009*). Paired-pulse inhibition of GABAergic IPSCs is significantly reduced in $App^{-/-}$ hippocampal slices (*Seabrook et al., 1999*). However, the molecular mechanism underlying this phenotype is not fully understood. Given that GABAergic function is commonly disturbed in many neuronal disorders including AD (*Braat and Kooy, 2015a*; *Verret et al., 2012*), identifying mechanisms whereby APP regulates GABAergic signaling and synaptic inhibition may provide a link between the endogenous function of APP and the etiology of AD.

The K$^+$-Cl$^-$ cotransporter, KCC2, is broadly expressed in neuronal membrane of the adult CNS (*Kaila et al., 2014*). KCC2 functions in setting the proper intracellular Cl$^-$ concentration ([Cl$^-$]$_i$) by transporting Cl$^-$ against the concentration gradient (*Ben-Ari, 2002*; *Ben-Ari et al., 2012*). KCC2 maintains low [Cl$^-$]$_i$ in mature neurons, which is essential for maintaining proper postsynaptic inhibition mediated by GABA$_A$ receptors (GABA$_A$Rs) and glycine receptors (GlyRs) (*Braat and Kooy, 2015a*; *Kaila et al., 2014*; *Rivera et al., 1999*). Impaired KCC2 activity and subsequent increases in [Cl$^-$]$_i$ occur in several neurological disorders (*Boulenguez et al., 2010*; *Coull et al., 2003*; *Tang et al., 2016*), leading to a depolarizing action of GABA$_A$R mediated currents due to a positive shift of E$_{GABA}$, and over-excitation of neuronal network activity (*Ben-Ari, 2002*; *Ben-Ari et al., 2012*; *Blaesse et al., 2009*). Recent experiments on KCC2 processing suggest that the intrinsic ion transport rate, cell surface stability, and trafficking of plasmalemmal KCC2 are rapidly and reversibly modulated by the (de)phosphorylation of critical serine and tyrosine residues at the C-terminus of this protein (*Lee et al., 2010*, *2007*). Increased tyrosine phosphorylation of KCC2 leads to enhanced degradation of KCC2 and thus reduced KCC2 protein levels (*Lee et al., 2011*, *2010*). Therefore, abundance of KCC2 protein and the E$_{GABA}$ of a neuron can be regulated by phosphorylation and

degradation of KCC2. However, which signaling pathways regulate these phosphorylation events and KCC2 degradation are largely unknown.

In the current study, we aimed to elucidate the synaptic mechanisms underlying APP regulation of hippocampal GABAergic inhibition. We show that APP mediates GABAergic inhibition via a direct protein-protein interaction with KCC2, which stabilizes KCC2 on the cell membrane. APP deficiency causes a loss of acting force holding KCC2 on site, resulting in increased KCC2 degradation via both tyrosine-phosphorylation and ubiquitination, decreased KCC2 levels and, sequentially, a depolarizing shift of $E_{GABA}$ and impairment in $GABA_AR$ activities.

## Results

### APP deficiency results in a depolarizing shift of $E_{GABA}$

In order to test the idea that loss of APP changes the cellular ionic balance of $Cl^-$, thereby altering GABAergic transmission, we used gramicidin perforated patch clamp measurements in acute hippocampal brain slices (*Figure 1A*) that allows maintenance of an intact intracellular chloride concentration to estimate the IV-relationship of GABAergic currents (*Chavas and Marty, 2003*). We observed a positive shift of $E_{GABA}$ in $App^{-/-}$ hippocampal CA1 compared to WT controls (WT, $-83 \pm 6$ mV;

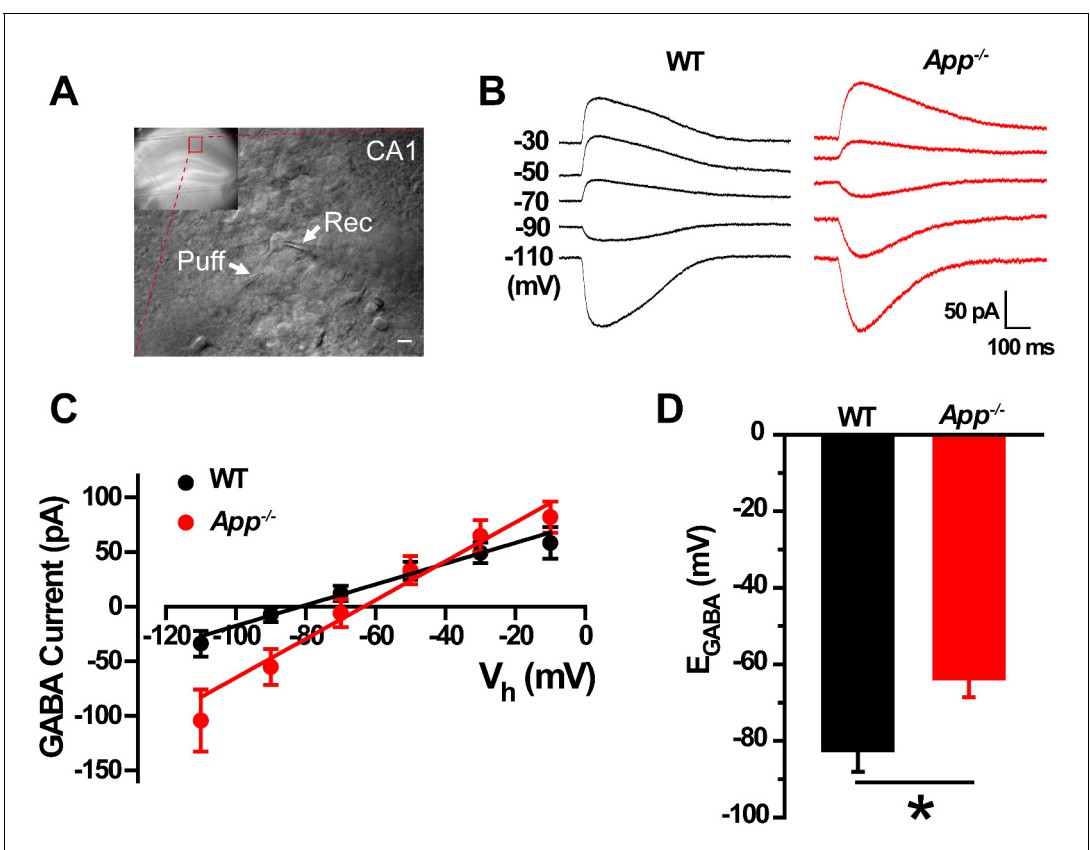

**Figure 1.** GABA reversal potential shifts toward depolarization in hippocampus of $App^{-/-}$ mice. (**A**) Slice preparations were made from the hippocampus and neurons were recorded in CA1 (inset). Pipettes were positioned to clamp neurons (Rec) while puffing molecules onto the cell (Puff). Scale bar: 10 μm. (**B**) Sample traces showing perforated patch recording of $E_{GABA}$ in WT and $App^{-/-}$ hippocampal slices. Currents were recorded at the indicated holding potentials shown to the left of each trace. (**C**) Graph shows *I–V* plots for the traces shown in B. (**D**) Quantification of $E_{GABA}$ shows a significant depolarizing shift in $App^{-/-}$ mice compared to WT controls (WT, n = 16 cells from five mice; $App^{-/-}$, n = 20 cells from five mice). *p<0.05; Student's *t*-test.

The following figure supplement is available for figure 1:

**Figure supplement 1.** GABA reversal potential shifts toward depolarization in $App^{-/-}$ hippocampal cultures.

$App^{-/-}$, $-64 \pm 5$ mV; p=0.02) (*Figure 1B–D*), accompanying with identical resting membrane potentials (WT, $-61 \pm 1.4$ mV; $App^{-/-}$, $-59 \pm 1.5$ mV; p=0.2). We repeated perforated patch recordings in hippocampal cultures and observed a similar depolarizing shift of $E_{GABA}$ in $App^{-/-}$ neuronal cultures (WT, $-82.5 \pm 4$ mV; $App^{-/-}$, $-67 \pm 3.8$ mV; p=0.02) (*Figure 1—figure supplement 1*).

## uIPSC amplitude and GABA$_A$R α1 subunit levels are reduced in $App^{-/-}$ hippocampus

Loss of APP impairs GABAergic synaptic transmission (*Seabrook et al., 1999*; *Yang et al., 2009*). Given that we see a change in $E_{GABA}$ in hippocampal neurons, we next asked whether loss of APP alters hippocampal IPSCs and GABA$_A$R expression. To do so, we performed dual whole-cell recordings in primary hippocampal cultures. We simultaneously evaluated electrically evoked presynaptic action potentials of a GABAergic interneuron labeled with GAD67$^{+/GFP}$, while patching a neighboring glutamatergic neuron and recording post-synaptic unitary IPSCs (uIPSCs) at a holding potential of $-60$ mV (*Figure 2A*). The mean uIPSC amplitude, evoked by a single presynaptic action potential in the GABAergic neuron, was significantly reduced in $App^{-/-}$ hippocampal cultures compared to WT control cultures (WT, $178 \pm 42$ pA; $App^{-/-}$, $80 \pm 19$ pA; p=0.03) (*Figure 2B–C*). Similarly, puffing the GABA$_A$R agonist isoguvacine significantly reduced the amplitude of inhibitory currents in $App^{-/-}$ neurons compared to WT neurons (WT, $245 \pm 26$ pA; $App^{-/-}$, $146 \pm 24$ pA; p=0.01) (*Figure 2F–G*). In contrast, the paired-pulse ratio (PPR) of uIPSCs in response to two consecutive action potentials in GABAergic neurons, at an inter-spike interval of 100 ms and 150 ms to assess synaptic release probability (*Zucker and Regehr, 2002*), was similar between WT and $App^{-/-}$ (*Figure 2D–E*) (100 ms; WT, $0.47 \pm 0.04$; $App^{-/-}$, $0.56 \pm 0.03$; p=0.2; 150 ms; WT, $0.51 \pm 0.06$; $App^{-/-}$, $0.55 \pm 0.05$; p=0.75). Consistent with impaired post-synaptic function accounting for these changes in APP mutants, we did not see a decrease in miniature IPSC (mIPSC) frequency, a measure of presynaptic efficacy (*Fiszman et al., 2005*; *Goswami et al., 2012*), when we compared $App^{-/-}$ and WT hippocampal slices (Frequency; WT, $2.9 \pm 0.8$ Hz; $App^{-/-}$ $2.6 \pm 0.7$ Hz; p=0.8) (*Figure 2—figure supplement 1C–D*). Meanwhile mIPSC amplitude remained unchanged as well (Amplitude; WT, $14.8 \pm 0.4$ pA; $App^{-/-}$, $13.9 \pm 0.8$ pA; p=0.7) (*Figure 2—figure supplement 1C–D*). In line with similar frequencies of mIPSCs in WT and $App^{-/-}$ hippocampus, we observed identical numbers of GABAergic neurons in the hippocampus of $App^{-/-}$ and WT mice (*Figure 2—figure supplement 1A–B*, *Figure 2—source data 1*). Finally, we looked at the abundance of GABA$_A$Rs in the hippocampus of APP mutant and WT animals. $App^{-/-}$ hippocampal lysates display reduced immunoreactivity of the α1 subunit, mediating fast inhibition, of the GABA$_A$R but not the other GABA$_A$R subunits, (*Figure 2H–I*, *Figure 2—source data 1*). Together, these data suggest that loss of APP may cause postsynaptic deficits in GABA$_A$R mediated GABAergic inhibition.

## APP deficiency leads to a reduction in total and plasma membrane KCC2 protein levels in hippocampus

The electrical response of GABA$_A$Rs depends on the Cl$^-$ equilibrium potential. The best characterized effectors of the Cl$^-$ distribution in the CNS are KCC2 and the Na-K-2Cl cotransporter (NKCC1) (*Ben-Ari, 2002*; *Kaila et al., 2014*; *Lee et al., 2011*; *Rivera et al., 1999*). Since neuronal [Cl$^-$]$_i$ is mainly determined by the opposing activities of the Cl$^-$ extruding transporter, KCC2, and the Cl$^-$ importing NKCC1 (*Blaesse et al., 2009*), we compared hippocampal KCC2 and NKCC1 levels in WT and $App^{-/-}$ mice. We found that KCC2, but not NKCC1, protein levels were significantly reduced in $App^{-/-}$ hippocampus compared to WT littermate controls (*Figure 3A–B*, *Figure 3—source data 1*). KCC2 is mainly localized to the cell surface (*Gauvain et al., 2011*). To determine whether plasma membrane localization of KCC2 is decreased in APP mutants, we used surface biotinylation in hippocampal tissue (*Figure 3A*) or HEK293 cells transfected with KCC2 alone or KCC2 and hAPP695 (*Figure 3C*). We observed significantly reduced plasma membrane KCC2 levels in both $App^{-/-}$ hippocampus and HEK293 cells without co-transfection of hAPP695 (*Figure 3B and D*, *Figure 3—source data 1*). To examine KCC2 levels in intact tissue, we performed immunohistochemical staining in slice with anti-KCC2 antibody. We observed a selective loss of KCC2 immunoreactivity in $App^{-/-}$ mice in the CA1 region of the hippocampus (*Figure 3E*). To determine if this is due to altered transcription of KCC2 in APP mutants, we compared hippocampal KCC2 mRNA levels in $App^{-/-}$ and WT controls. KCC2 mRNA abundance was similar in $App^{-/-}$ and WT hippocampus (*Figure 3F*, *Figure 3—*

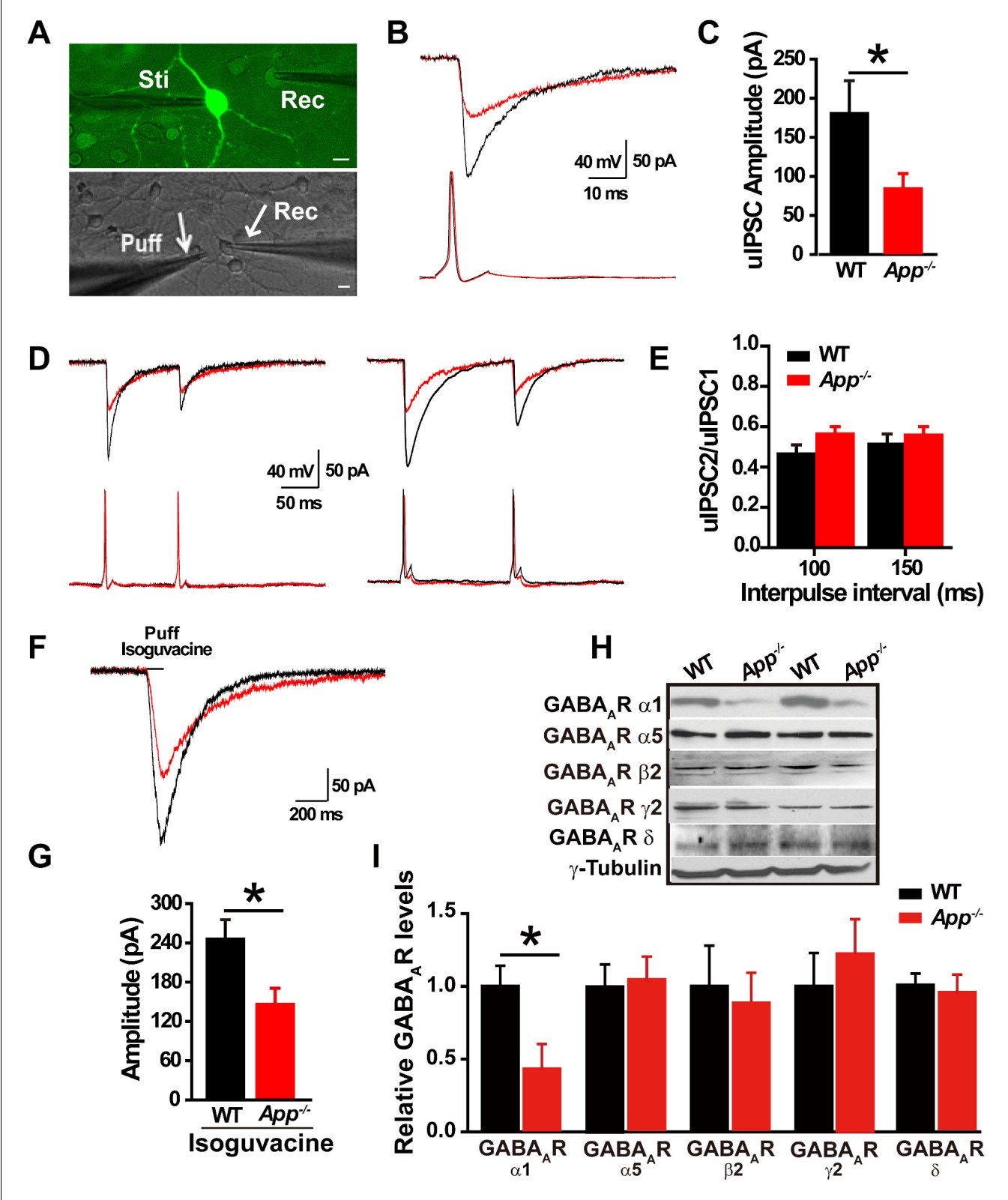

**Figure 2.** Decreased inhibitory IPSC amplitude and GABA$_A$R α1 protein levels in *App*$^{-/-}$ mice. (**A**) Images of 12–14 DIV hippocampal cultures of *Gad67*$^{+/GFP}$ mice. In the upper panel, under fluorescent illumination, GABAergic neurons appear green, and were stimulated (Sti) while recording from nearby glutamatergic cells (Rec) and puffing solutions (puff, lower panel DIC). Scale bar: 10 μm. (**B**) Sample traces showing postsynaptic uIPSC responding to presynaptic action potentials induced by short depolarizing voltage pulse injection (2 ms) to GABAergic neurons. (WT, black line; *App*$^{-/-}$, red line). (**C**)
*Figure 2 continued on next page*

*Figure 2 continued*

Quantification of uIPSC amplitude shows a significant decrease in $App^{-/-}$ mice. (WT, n = 18 cells from five mice; $App^{-/-}$, n = 17 cells from five mice). (D) Sample traces showing uIPSC recordings responding to injection of paired pulses to presynaptic GABAergic neurons (100 ms interpulse interval, left trace and 150 ms interpulse interval, right trace). (E) Quantification of paired pulse ratio (PPR) of uIPSCs with 100 ms and 150 ms interpulse intervals shows no significant difference between genotypes. (F) Sample traces showing evoked inhibitory currents responses to puffing 100 μM isoguvacine. (G) Quantification of isoguvacine-evoked inhibitory current amplitudes shows a significant decrease in $App^{-/-}$ mice. (WT, n = 14 cells from three mice; $App^{-/-}$, n = 14 cells from three mice). (H) Representative immunoblots of hippocampal extracts from WT and $App^{-/-}$ littermates. (I) Quantification of the immunoblots reveals a significant decrease of GABA$_A$R α1, but not other GABA$_A$R subunits, levels in $App^{-/-}$ mice. Representative immunoblots of western blotting were from single experiment using three pairs hippocampal lysates, two repeats. *p<0.05; Student's *t*-test.

The following source data and figure supplements are available for figure 2:

**Source data 1.** Contains source data for *Figure 2* and all accompanying *Figure 2—figure supplements 1,2*.
**Figure supplement 1.** Similar numbers of GABAergic interneurons and mIPSC in WT and $App^{-/-}$ hippocampus.
**Figure supplement 2.** Identical mEPSC and GluRs levels in between WT and $App^{-/-}$ hippocampus.

*source data 1*). Together, these results indicate that APP is required to maintain normal KCC2 protein levels in the hippocampus via a post-transcriptional mechanism.

## Full length APP stabilizes KCC2

To determine whether APP could stabilize KCC2 protein, we co-transfected HEK293 cells with KCC2 and hAPP695 cDNAs and assayed subsequent KCC2 protein levels. Cells co-transfected with both constructs displayed higher levels of KCC2 protein compared to cells transfected with only KCC2 (*Figure 3G*). APP is proteolytically processed into multiple extracellular and intracellular fragments. We tested these fragments individually to determine which fragment conferred KCC2 stability. We transfected HEK293 cells with KCC2+C99, KCC2+sAPPβ, or KCC2+hAPP695. Only full length APP but not the intracellular or extracellular fragments of APP prevented KCC2 degradation (*Figure 3H*, *Figure 3—source data 1*). Furthermore, using a sAppβ knock in mouse model (sAppβKi), we observed similarly reduced KCC2 levels in sAppβKi hippocampus compared to WT mice (*Figure 3I–J*, *Figure 3—source data 1*). Together, our results indicate that full length APP is required to maintain normal levels of KCC2 protein.

## APP and KCC2 physically interact to limit tyrosine phosphorylation and thus degradation of KCC2

To test whether APP regulates KCC2 through direct protein-protein interaction, we performed immunoprecipitation of KCC2 in HEK293 cells co-transfected with constructs encoding KCC2 and hAPP695. First, Coomassie blue staining was performed in transfected HEK293 cells showing a distribution of all proteins, including APP, pulled down by anti-KCC2 antibody (*Figure 4—figure supplement 1*). Next, APP was detected in anti-KCC2 immunoprecipitates (IP). Similarly, immunoprecipitation of APP co-precipitated KCC2 (*Figure 4A*). However, immunoprecipitation of the GABA$_A$R α1 subunit co-transfected with hAPP695, we did not observe a direct physical interaction between APP and GABA$_A$R α1 subunit (*Figure 4B*). Immunoprecipitation of KCC2 using hippocampal lysates co-precipitated APP as well (*Figure 4C*). Further, the APP-KCC2 interaction result was strengthened by proximity ligation assay (PLA) which would sensitively detect a very close surface interaction between proteins (*Söderberg et al., 2006*) (*Figure 4D*).

KCC2 is regulated post-translationally by tyrosine phosphorylation (*Lee et al., 2010*; *Wake et al., 2007*). Increased tyrosine phosphorylation of KCC2 enhances lysozomal degradation and thus decreases KCC2 levels (*Lee et al., 2011*, *2010*). We hypothesized that APP may normally interact with KCC2 to prevent tyrosine phosphorylation and subsequent degradation. To test this hypothesis, we examined whether tyrosine phosphorylated KCC2 levels increased in the absence of APP. Hippocampal tissue from $App^{-/-}$ and WT mice was lysed and immunoprecipitated with anti-KCC2 antibody (*Lee et al., 2010*). Precipitates were blotted for tyrosine phosphorylated proteins using the anti-P-Tyr antibody, 4G10. We observed a robust increase in tyrosine phosphorylated KCC2 levels in APP

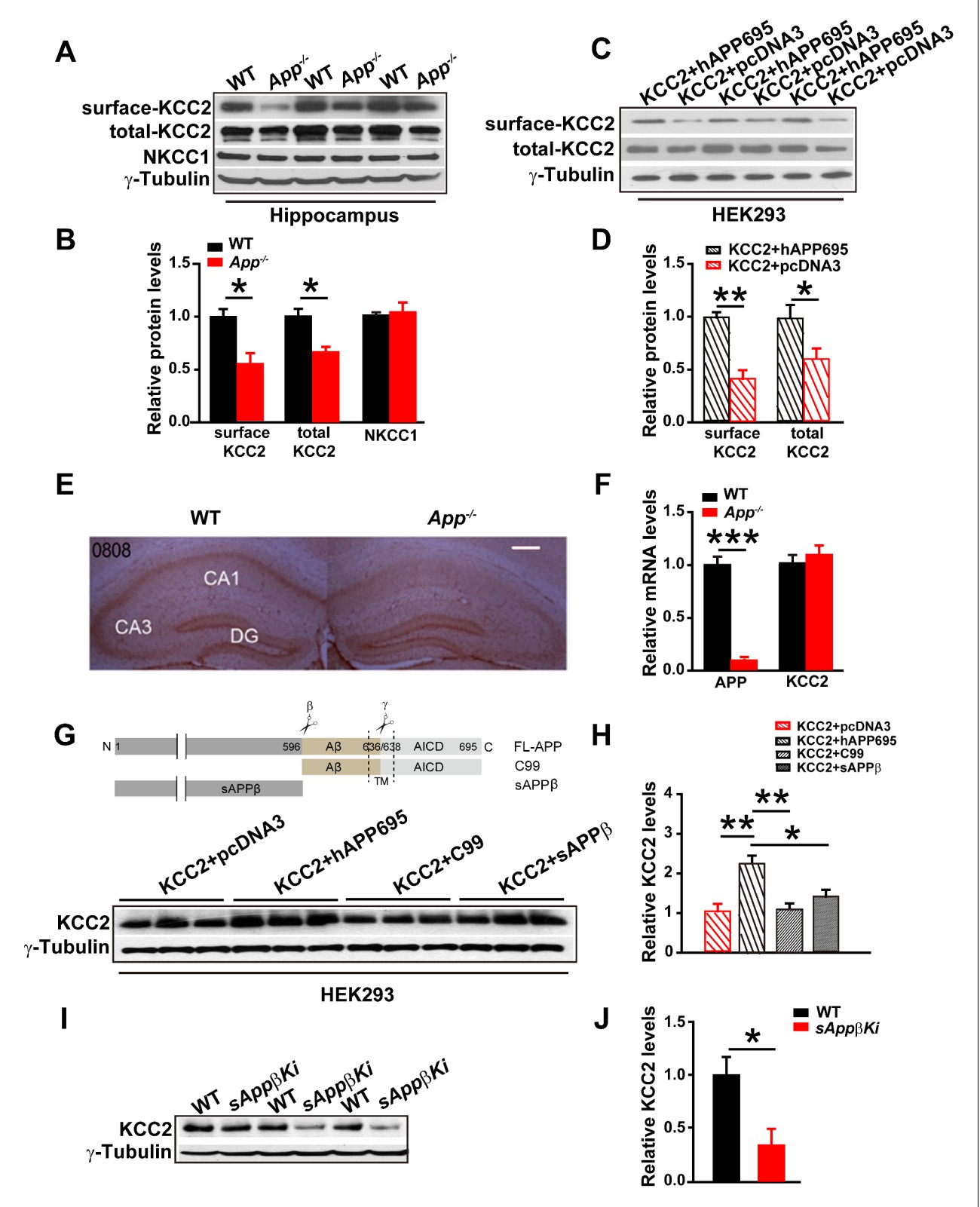

**Figure 3.** Full length APP is required for maintaining normal total and surface KCC2 protein levels. (**A**) Representative immunoblots of hippocampal extracts from WT and *App*[-/-] littermates. (**B**) Quantification of the immunoblots reveals a significant decrease in surface and total KCC2 protein levels in *App*[-/-] mice, while NKCC1 expression levels remain unchanged. Representative immunoblots of western blotting were from single experiment using three pairs of hippocampal lysates, three independent experiments (**C**) Representative immunoblots of HEK293 cells transfected with KCC2 and

*Figure 3 continued on next page*

*Figure 3 continued*

hAPP695. (D) Quantification shows surface and total KCC2 protein levels are elevated when HEK293 cells are transfected with both constructs. (E) DAB staining for KCC2 in hippocampal sections from *App*$^{-/-}$ and WT littermates shows decreased KCC2 levels in CA1 of hippocampus in *App*$^{-/-}$ mice. Scale bar: 400 µm. (F) qRT-PCR displays similar KCC2 mRNA levels in WT and *App*$^{-/-}$ hippocampus. APP mRNA serves as a positive control. Each value represents the mean±SEM of at least three samples per genotype. (G) Representative immunoblots of transfected HEK293 cells. (H) Quantification of the immunoblots reveals an increase in KCC2 levels when we co-transfect hAPP695, but not when co-transfect with C99 or with sAPP*β*. (I) Representative immunoblots of hippocampal extracts from *sAppβKi* and WT littermates. (J) Quantification of the immunoblots reveals a significant decrease of KCC2 protein levels in *sAppβKi* mice. Representative immunoblots were from single experiment using three pairs of HEK293 cells/ or hippocampal lysates, two repeats. *p<0.05; **p<0.01; **p<0.001; Student's *t*-test.

The following source data is available for figure 3:

**Source data 1.** Contains source data for *Figure 3*.

mutants compared to littermate controls (*Figure 4E*). Moreover, total KCC2 protein levels remained decreased in *App*$^{-/-}$ (*Figure 4E*) (Relative IOD: WT, 544.86; *App*$^{-/-}$, 412.56. hippocampal lysates of three mice per genotype) indicating an increase in tyrosine phosphorylation of KCC2 accompanies the reduction in total KCC2 levels. To further demonstrate that KCC2 instability is due, at least in part, to phosphorylation by a tyrosine kinase in the absence of APP, we tested whether blockade of tyrosine kinases by PP2, a potent inhibitor of Src-family tyrosine kinases (*Bi et al., 2000*; *Lee et al., 2007*), enhances KCC2 protein levels. Treatment of HEK293 cells transfected with KCC2 with PP2 at a concentration of 20 µM for 30 min significantly increased KCC2 levels compared to vehicle controls (*Figure 4F–G*, *Figure 4—source data 1*). Together, these results suggest that APP-KCC2 interaction functions, at least in part, to limit tyrosine phosphorylation of KCC2, loss of APP leads to excessive tyrosine phosphorylation and premature degradation of KCC2, resulting in reduced KCC2 protein levels in APP mutants.

## KCC2 abundance and function can be rescued by CLPs

Next, we tested whether restoring KCC2 function in APP mutant mice would rescue changes in $E_{GABA}$ in hippocampal neurons. Chloride extrusion enhancers, CLP257 and CLP290, have been recently shown to restore Cl$^-$ extrusion capacity only in neurons with reduced KCC2 activity (*Gagnon et al., 2013*). We incubated hippocampal slices from APP mutant mice with CLP257 at 100 µM for 2 hr, and compared $E_{GABA}$ in slices treated with CLP257 or vehicle. Activation of KCC2 in *App*$^{-/-}$ hippocampus resulted in a hyperpolarizing shift of $E_{GABA}$ (CLP257, −66 ± 3 mV; Control, −54 ± 2 mV; p=0.03) (*Figure 5A–B*, *Figure 5—source data 1*). We next tested whether restoration of Cl$^-$ extrusion by chronic treatment with CLP290 (*Gagnon et al., 2013*) rescues KCC2 expression levels. CLP290 or vehicle was intraperitoneally injected once daily for seven consecutive days in *App*$^{-/-}$ mice, and hippocampal KCC2 levels were analyzed by western blotting. We observed a significant increase in KCC2 levels in CLP290 treated animals compared to vehicle treated *App*$^{-/-}$ mice (*Figure 5C–D*, *Figure 5—source data 1*).

It has been proposed that CLPs might increase KCC2 levels by reducing membrane KCC2 turnover (*Gagnon et al., 2013*). We then examined, in 293 cells transfected with KCC2, if treatment with CLP257 would indeed increase levels of membrane KCC2, and observed significantly enhanced surface KCC2 levels in CLP257 treated cells compared to vehicle control (*Figure 5E–F*, *Figure 5—source data 1*). These experiments could not distinguish whether CLP restored KCC2 levels by increasing insertion or decreasing removal of KCC2. Instead, the study suggested a rescue of KCC2 surface levels and function by CLPs in *App*$^{-/-}$ mice.

## GABA$_A$R α1 subunit abundance and GABA$_A$R mediated inhibition are dependent on KCC2

We observed decreases in GABA$_A$R α1 subunit protein levels and smaller evoked IPSC amplitudes in APP mutants (*Figure 2*). We then tested whether potentiating KCC2 function would rescue GABA$_A$R mediated inhibitory currents in APP mutant hippocampal slices. Evoked whole-cell inhibitory currents, induced by puffing the GABA$_A$R agonist isoguvacine, were recorded after incubation with CLP257 or vehicle for 2 hr, respectively. Evoked inhibitory current amplitudes were significantly

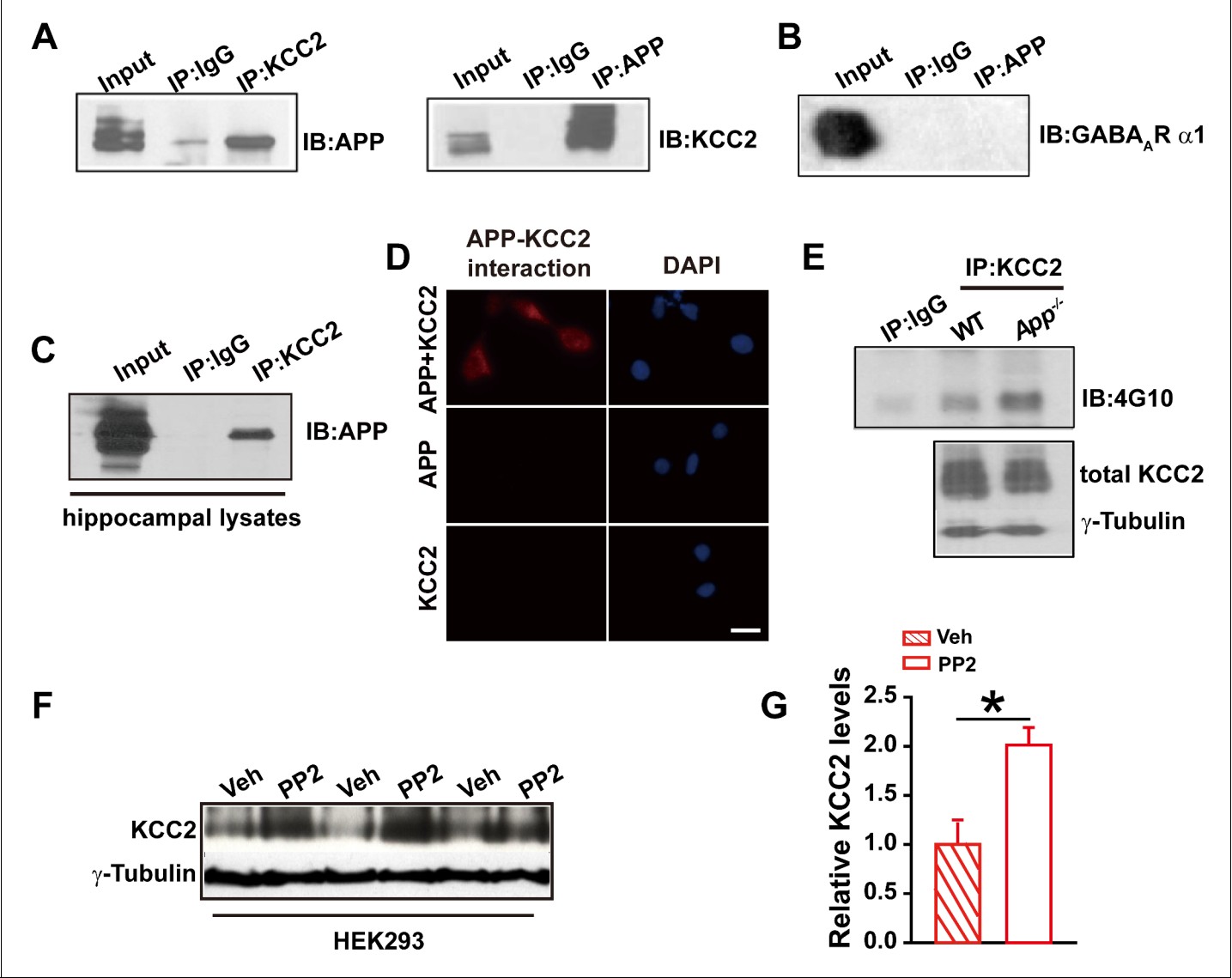

**Figure 4.** APP interacts with KCC2 to limit KCC2 tyrosine phosphorylation and degradation. (A) KCC2 interacts with APP in HEK293 cells by co-IP with both KCC2 and APP antibodies. Rabbit IgG (IP: IgG) was used as a negative control. (B) GABA$_A$R α1 subunit does not coimmunoprecipitate with APP. (C) KCC2 interacts with APP by co-IP in hippocampal lysates. (D) KCC2 interacts with APP by proximity ligation assay (PLA). HEK293 cell was transfected with APP and KCC2, APP alone or KCC2 alone. Red: PLA signal indicates an existence of APP-KCC2 interaction; Blue: DAPI. Scale bar: 20 μm. (E) Hippocampal extracts from WT and *App*$^{-/-}$ littermates were immunoprecipitated with an anti-KCC2 antibody (IP: KCC2) and probed with 4G10 antibody (anti-Phospho-tyrosine). Tyrosine phosphorylation of KCC2 is increased in *App*$^{-/-}$ mice. Total KCC2 levels are also decreased in *App*$^{-/-}$ mice. (Extracts were from three mice per genotype). (F) Representative immunoblots of KCC2 transfected HEK293 cells incubated with PP2 or vehicle. (G) Quantification of the immunoblots reveals a significant increase in KCC2 protein levels in PP2 treated HEK293 cells. Representative immunoblots of western blotting were from single experiment using three pairs of HEK293 lysates, two repeats. *p<0.05; Student's *t*-test.

The following source data and figure supplement are available for figure 4:

**Source data 1.** Contains source data for *Figure 4*.
**Figure supplement 1.** Coomassie-stained SDS-PAGE gel showing KCC2 binding proteins after immunoprecipitated with an anti-KCC2 antibody (IP: KCC2).

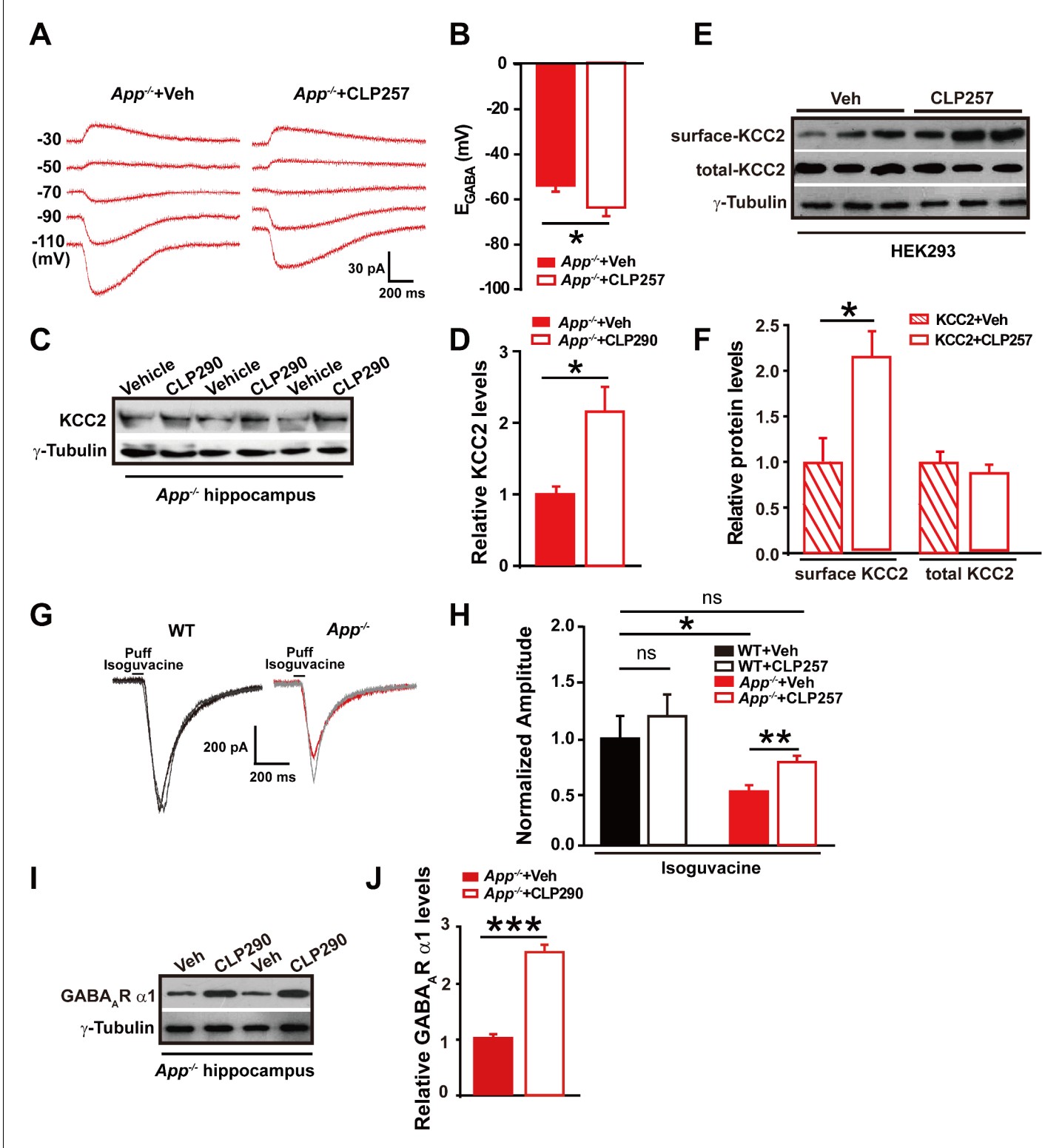

**Figure 5.** Restoring KCC2 expression and function in $App^{-/-}$ mice rescues hippocampal $E_{GABA}$, $GABA_A R$ expression and IPSC amplitude. (A) Sample traces showing perforated patch recording of $E_{GABA}$ in CLP257 or vehicle treated $App^{-/-}$ hippocampal slices. Currents were recorded at the indicated holding potentials shown to the left of each trace. (B) Quantification of $E_{GABA}$ shows a significant hyperpolarizing shift in $App^{-/-}$ in CLP257 treated slices compared to vehicle controls (CLP257, n = 5 cells from three mice; Control, n = 5 cells from three mice). (C) Representative immunoblots of hippocampal extracts from $App^{-/-}$ treated with CLP290 or vehicle. (D) Quantification of the immunoblots reveals increased KCC2 protein levels in CLP290 treated $App^{-/-}$ mice. (E) Representative immunoblots of HEK293 cells transfected with KCC2 incubated with CLP257 or vehicle. (F)
*Figure 5 continued on next page*

*Figure 5 continued*

Quantification of the immunoblots reveals a significant increase in surface KCC2 protein levels when treated with CLP257, while total KCC2 levels remain unchanged, suggesting CLP257 might regulate KCC2 activity through blocking the latter's turnover. (G) Sample traces of evoked inhibitory currents in response to puffs of 100 μM isoguvacine. (H) Quantification of isoguvacine-evoked inhibitory current amplitudes shows a significant increase in CLP257 treated $App^{-/-}$ hippocampus compared to vehicle controls (CLP257, n = 15 cells from three mice; Control, n = 15 cells from three mice), but no difference between CLP257 treated WT hippocampus and vehicle treated controls (CLP257, n = 9 cells from three mice; Control, n = 11 cells from three mice), suggesting that restoring KCC2 function rescues GABA$_A$R mediated responses. (I) Representative immunoblots of hippocampal extracts from $App^{-/-}$ treated with CLP290 or vehicle. (J) Quantification of the immunoblots reveals increased GABA$_A$R α1 protein levels in CLP290 treated $App^{-/-}$ mice. Representative immunoblots of western blotting were from single experiment using three pairs of HEK293 cells/ or hippocampal lysates of two independent experiments. *p<0.05; **p<0.01; ***p<0.001; Student's *t*-test; quantified data of H, two-way ANOVA with post hoc tests.

The following source data is available for figure 5:

**Source data 1.** Contains source data for *Figure 5*.

larger in CLP257 treated $App^{-/-}$ slices compared to vehicle controls ($App^{-/-}$: CLP257, 410 ± 34 pA; Vehicle, 278 ± 31 pA; p=0.008) (*Figure 5G–H*). Importantly, CLP257 treated WT slices did not differ significantly compared to vehicle controls (WT: CLP257, 612 ± 99 pA; Vehicle, 510 ± 109 pA; p=0.5) (*Figure 5G–H*). Additionally, chronic treatment with CLP290 rescued GABA$_A$R α1 subunit protein levels as well (*Figure 5I–J*, *Figure 5—source data 1*). These results support a model in which APP loss of function results in reduced KCC2 levels and decreased KCC2 function, which causes the attenuation of GABA$_A$R α1 expression and GABA$_A$R mediated inhibitory current amplitudes, leading to a decrease of inhibitory tone in the hippocampus of APP mutants.

To test whether GABA$_A$R α1 level changes are dependent on the presence of KCC2, we transfected HEK293 cells with GABA$_A$R α1 or β2 subunits with and without co-transfection of KCC2. Interestingly, we only observed significantly reduced GABA$_A$R α1, but not β2, subunit levels in the absence of KCC2 (*Figure 6A–B*, *Figure 6—source data 1*), suggesting a selective role of KCC2 on GABA$_A$R α1 expression.

The mechanisms underlying the actions of KCC2 on GABAergic synapses are currently unclear (*Chudotvorova et al., 2005*). We next conducted experiments to elucidate how KCC2 regulates GABA$_A$R α1 expression and thus GABA$_A$R mediated inhibition. During immunoprecipitation of the GABA$_A$R α1 subunit co-transfected with KCC2, we did not observe a direct physical interaction between KCC2 and GABA$_A$R α1 subunit (*Figure 6C*), indicating that KCC2 regulation of GABA$_A$R α1 expression was unlikely through a direct protein-protein interaction.

In a previous report the ubiquitin-like protein PLIC-1, which regulates the membrane trafficking of GABA$_A$R, was shown to directly interact with GABA$_A$Rs and promote their accumulation at the cell surface (*Saliba et al., 2008*). We next examined if KCC2 regulates GABA$_A$R α1 levels through affecting PLIC-1 expression, however, Western blotting result showed identical PLIC-1 levels in hippocampus of WT and $App^{-/-}$ mice (*Figure 6D–E*, *Figure 6—source data 1*).

It has been shown that expression of GABA$_A$R subunits is regulated by KCC2 expression through affecting intracellular Cl$^-$ gradient (*Succol et al., 2012*). We thus proposed that KCC2-mediated Cl$^-$ extrusion might underlie KCC2 regulation of GABA$_A$R α1 protein levels. If this was the case, treatment of WT hippocampus with a KCC2 inhibitor, VU02450551, would result in a reduction of α1 subunit of GABA$_A$R protein levels as seen in $App^{-/-}$ hippocampus. Hippocampal GABA$_A$R α1, several other GABA$_A$R subunits and GluRs levels were analyzed by western blotting after incubation with VU02450551 or vehicle for 2 hr in WT hippocampus slices. We observed a significant decrease in GABA$_A$R α1 levels, but not that of other GABA$_A$Rs and GluRs subunits in VU02450551 treated WT hippocampus compared to vehicle control (*Figure 6F and H*, *Figure 6—source data 1*). Moreover, incubation of $App^{-/-}$ hippocampus with KCC2 inhibitor, VU02450551, further reduced levels of GABA$_A$R α1 subunit but left GluR1 levels unchanged (*Figure 6G and H*, *Figure 6—source data 1*). These experiments suggested that blockade of Cl$^-$ extrusion by VU02450551 selectively affected levels of GABA$_A$R α1, but not other GABA$_A$R subunits tested in the present study.

It was well known that the expression of KCC2 and GABAergic inhibition parallel neuronal maturation and the emergence of low intracellular Cl$^-$ (*Ben-Ari et al., 2012*; *Blaesse et al., 2009*; *Rivera et al., 1999*). KCC2 exports Cl$^-$ and is weakly expressed at birth and upregulated as the brain

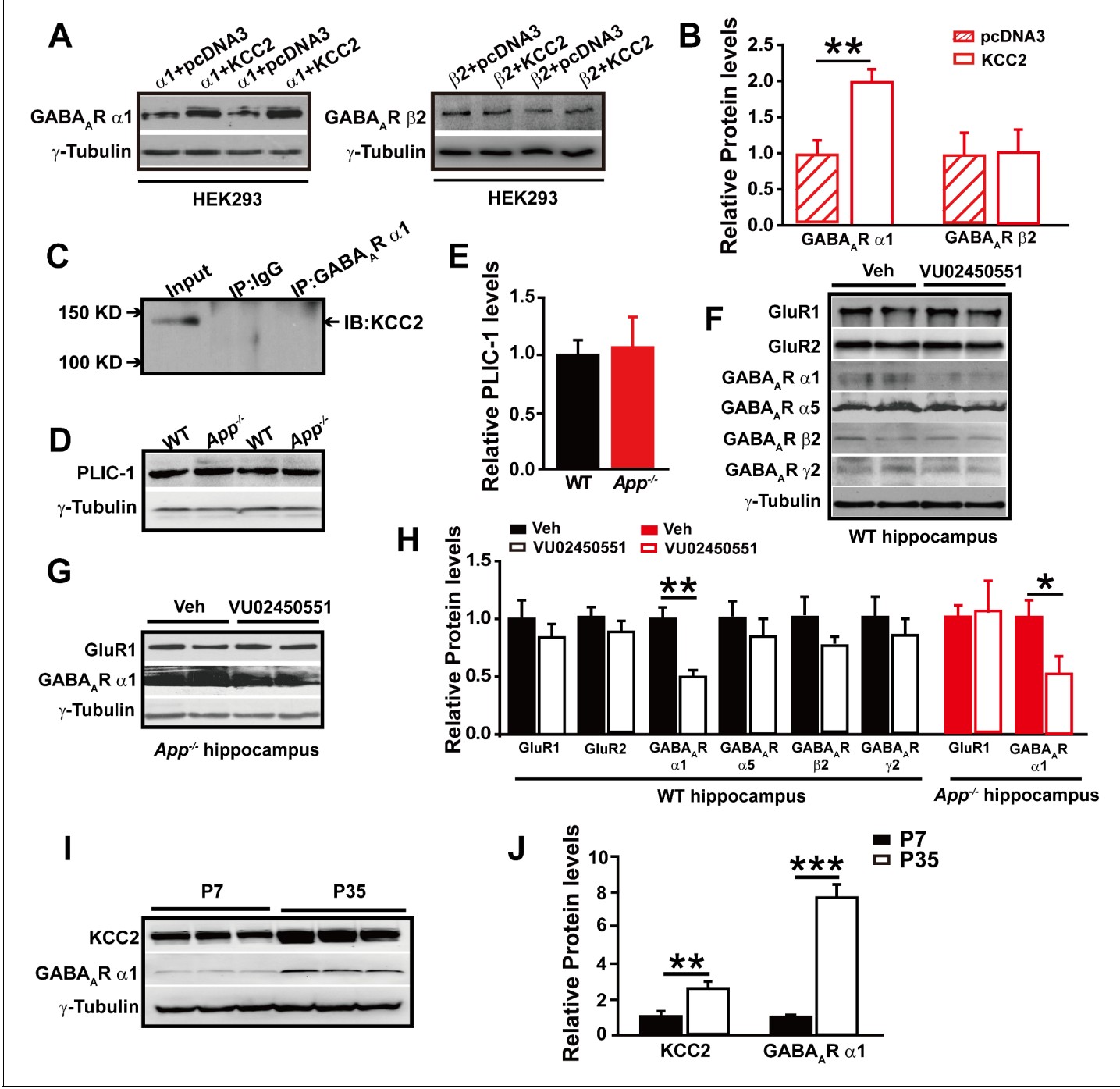

**Figure 6.** GABA$_A$R α1 subunit is regulated by KCC2 expression through affecting intracellular Cl$^-$. (**A**) Representative immunoblots of GABA$_A$R α1+KCC2, GABA$_A$R α1+pcDNA3, GABA$_A$R β2+KCC2, GABA$_A$R β2+pcDNA3, transfected HEK293 cells blotted with anti-GABA$_A$R antibodies. (**B**) Quantification of the immunoblots reveals a significant increase in GABA$_A$R α1 levels in HEK293 cells co-transfected with GABA$_A$R α1 and KCC2, indicating KCC2 positively regulates expression of GABA$_A$R α1. (**C**) GABA$_A$R-α1 does not co-immunoprecipitate with KCC2, suggesting that there was no protein-protein interaction between KCC2 and GABA$_A$R α1. (**D**) Representative immunoblots of hippocampal extracts from WT and $App^{-/-}$ littermates. (**E**) Quantification of the immunoblots reveals identical PLIC-1 levels in between WT and $App^{-/-}$ hippocampus. (**F**) Representative immunoblots of WT hippocampal slices incubated with a KCC2 inhibitor, VU02450551, or vehicle. (**G**) Representative immunoblots of $App^{-/-}$ hippocampal slices incubated with a KCC2 inhibitor, VU02450551, or vehicle. (**H**) Quantification of the immunoblots reveals a significant decrease only in GABA$_A$R α1 levels of hippocampal tissue treated with VU02450551. (**I**) Representative immunoblots of hippocampal extracts from P7 and P35 WT mice. (**J**) Quantification of the immunoblots reveals significantly increased KCC2 and GABA$_A$R α1 levels in P35 compared to P7 WT mice.

*Figure 6 continued on next page*

*Figure 6 continued*

Representative immunoblots of western blotting were from single experiment using three pairs of HEK293 cells/ or hippocampal lysates, two repeats. *p<0.05; **p<0.01; ***p<0.001; Student's *t*-test.

The following source data and figure supplement are available for figure 6:
**Source data 1.** Contains source data for *Figure 6*.
**Figure supplement 1.** Immunofluorescent staining of hippocampal KCC2 in P14 and P35 WT mice shows increased KCC2 levels in P35 WT mice.

matures (*Plotkin et al., 1997*; *Rivera et al., 1999*). To further evaluate the notion that rescue of GABA$_A$R $\alpha$1 expression and GABA$_A$R mediated inhibition in *App*$^{-/-}$ hippocampus by restoring KCC2 levels involves alteration in intracellular Cl$^-$, we compared hippocampal KCC2 and GABA$_A$R $\alpha$1 levels by western blotting in between postnatal day 7 (P7) (with high intracellular Cl$^-$ concentration) and P35 (with low intracellular Cl$^-$ concentration) WT mice, and observed significantly increased GABA$_A$R and KCC2 levels in P35 compared to P7 hippocampus (*Figure 6I–J*, *Figure 6—source data 1*). Similar to western blotting, immunofluorescent staining in P14 and P35 hippocampus displayed higher KCC2 levels in P35 compared to P14 mice (*Figure 6—figure supplement 1*). Data thus far suggested that rescue of GABA$_A$R $\alpha$1 levels by restoring KCC2 in *App*$^{-/-}$ hippocampus involved, at least in part, an alteration of KCC2 mediated Cl$^-$ extrusion.

## APP also functions to suppress ubiquitination and thus degradation of KCC2

Having shown that one way for APP to regulate KCC2 levels was to limit tyrosine phosphorylation and sequential degradation of KCC2, we speculated that blockade of tyrosine phosphorylation of KCC2 by mutating Y903A and Y1087A (mKCC2) which are two tyrosine phosphorylating sites of KCC2 may restore KCC2 levels in the absence of APP. However, mKCC2 levels were significantly lower without hAPP695 than with (*Figure 7A–B*, *Figure 7—source data 1*). Meanwhile, mKCC2 levels are identical in PP2 treated and non-treated HEK293 cells (*Figure 7C–D*, *Figure 7—source data 1*) confirming that mKCC2 did not undergo tyrosine phosphorylation, which met desired effect. Moreover, both IP and PLA examinations displayed direct protein-protein interaction between APP and mKCC2 (*Figure 7E–F*), suggesting that mutation of KCC2 at tyrosine phosphorylating sites did not disturb APP-KCC2 interaction. We then hypothesized that APP deficiency might cause a loss of acting force to hold mKCC2 on site leading to an increase in mKCC2 internalization, resulting in enhanced mKCC2 degradation via an unknown pathway rather than tyrosine phosphorylation. Interestingly, similar to co-transfection of mKCC2 with hAPP695 constructs, HEK293 cells co-transfected with mKCC2 and cylindromatosis (CYLD, a deubiquitinating enzyme) displayed higher levels of mKCC2 protein levels compared to cells transfected with only mKCC2 (*Figure 7G–H*, *Figure 7—source data 1*), implicating that mKCC2 might undergo ubiquitination and subsequent degradation in the absence of APP.

Next, treating mKCC2 transfected HEK293 cells (in the presence or absence of APP co-transfection) with the ubiquitination inhibitor MG132, we observed that MG132 increased levels of mKCC2 only in the absence (*Figure 8A left–B*, *Figure 8—source data 1*) but not in the presence (*Figure 8A right–B*, *Figure 8—source data 1*) of hAPP695, supporting that APP functions to limit mKCC2 ubiquitination. Moreover, using MG132 to block ubiquitination in HEK293 cells transfected with KCC2, we confirmed that ubiquitination occurred in KCC2 as well in the absence of APP (*Figure 8C–D*, *Figure 8—source data 1*).

We then examined whether KCC2 ubiquitination contributed to reduction of KCC2 levels in *App*$^{-/-}$ hippocampus. Hippocampal tissue from *App*$^{-/-}$ and WT mice was lysed and immunoprecipitated with anti-KCC2 antibody. Precipitates were blotted for ubiquitin/ubiquitinated proteins using the anti-ubiquitin antibody. We observed a robust increase in ubiquitinated KCC2 levels in *App*$^{-/-}$ hippocampus compared to littermate controls (*Figure 8E–F*). Furthermore, incubating hippocampus containing *App*$^{-/-}$ slices with MG132 significantly increased KCC2 levels (*Figure 8G–H*, *Figure 8—source data 1*). Moreover, surface KCC2 levels could also be increased by MG132 blockade of

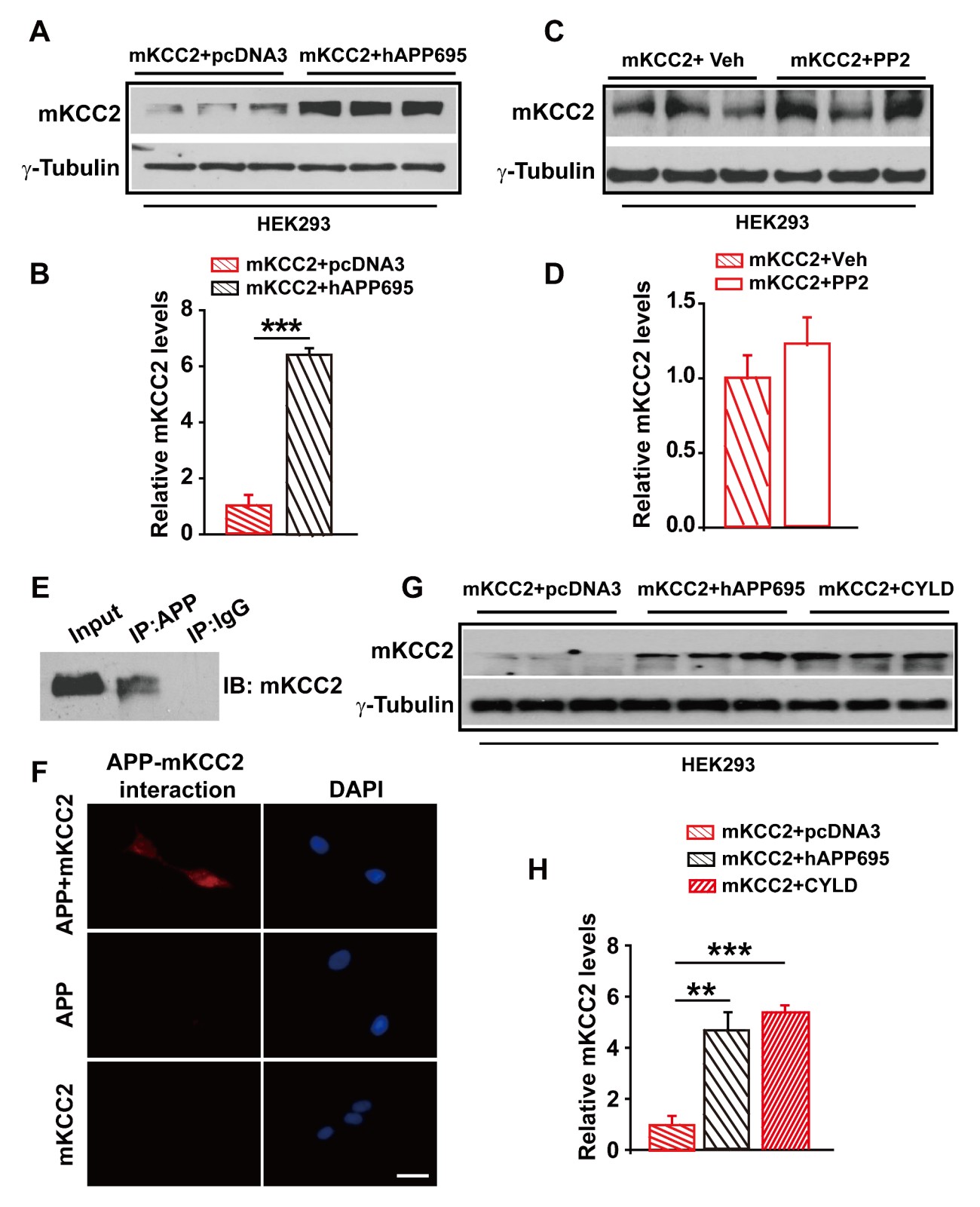

**Figure 7.** APP interacts with mKCC2 to limit mKCC2 ubiquitination and degradation. (**A**) Representative immunoblots of HEK293 cells transfected with mKCC2 and hAPP695. (**B**) KCC2 protein levels are elevated when HEK293 cells are transfected with both constructs. (**C**) Representative immunoblots of mKCC2 transfected HEK293 cells incubated with PP2 or vehicle. (**D**) Quantification of the immunoblots reveals identical KCC2 protein levels in between PP2 treated HEK293 cells and vehicle control. (**E**) mKCC2 interacts with APP. Rabbit IgG (IP: IgG) was used as a negative control. (**F**) PLA shows that

*Figure 7 continued on next page*

*Figure 7 continued*

mKCC2 interacts with as well. HEK293 cell were transfected with both APP and mKCC2, APP alone, or mKCC2 alone. Red: PLA signal indicates an existence of APP-mKCC2 interaction; Blue: DAPI. Scale bar: 20 μm. (G) Representative immunoblots of transfected HEK293 cells. (H) Quantification of the immunoblots reveals a significantly increased KCC2 protein levels in HEK293 cell transfected with hAPP695 and CYLD, respectively. Representative immunoblots of western blotting were from single experiment using three pairs of lysates, two repeats. **p<0.01; ***p<0.001; Student's *t*-test.

The following source data is available for figure 7:

**Source data 1.** Contains source data for *Figure 7*.

KCC2 ubiquitination in HEK293 cells and *App*$^{-/-}$ hippocampus (*Figure 8I–J*, *Figure 8—source data 1*).

Data thus far indicated that KCC2 levels could be regulated post-translationally by ubiquitination as well as tyrosine phosphorylation in the absence of APP. APP-KCC2 interaction played a crucial role in suppressing KCC2 internalization and subsequent degradation via both tyrosine phosphorylation and ubiquitination. APP deficiency resulted in a loss of acting force to hold membrane KCC2, leading to enhanced KCC2 degradation in the hippocampus.

## Restoring KCC2 by CLP257 did not rescue abnormal tonic GABAergic current in CA1 of *App*$^{-/-}$ mice

GABA has not only phasic but also tonic mode of action (tonic GABA current) and the latter involves extrasynaptic GABA$_A$Rs and/or reactive astrocytic GABA (*Curia et al., 2009*; *Jo et al., 2014*), and tonic GABA action has been implicated in AD mouse models (*Jo et al., 2014*; *Wu et al., 2014*; *Yarishkin et al., 2015*). We have previously shown that tonic GABA currents were impaired in the dentate gyrus (DG) of *App*$^{-/-}$ mice attributing to abnormal genesis and survival of DG newborn neurons (*Wang et al., 2014b*). We thus evaluated whether abnormal tonic GABA currents also occurred in APP deficient CA1 and, observed significantly reduced amplitude of tonic GABA current in the CA1 of *App*$^{-/-}$ mice compared to WT controls (WT, 19.1 ± 2.7 pA; *App*$^{-/-}$, 9.1 ± 0.7 pA; p=0.002) (*Figure 9—source data 1*). Interestingly, restoring KCC2 by CLP257 incubation of *App*$^{-/-}$ hippocampal slice did not rescue impaired tonic GABA current (*App*$^{-/-}$, 9.1 ± 0.7 pA; *App*$^{-/-}$+CLP257, 11.9 ± 1.6 pA; p=0.1) (*Figure 9—source data 1*), suggesting that aberrant KCC2 levels and function might underlie impaired phasic but not tonic GABA current in the CA1 of *App*$^{-/-}$ mice.

## Discussion

In this study, we demonstrate that APP regulation of KCC2 expression and function is required to maintain appropriate GABAergic signaling capacity in the hippocampus as summarized in *Figure 10*. This is supported by several lines of evidence: (1) APP deficiency results in a depolarizing shift of E$_{GABA}$ along with reduced amplitude of uIPSCs and tonic GABA current, (2) APP deficiency results in a significant reduction in the abundance of hippocampal KCC2, but not NKCC1, and GABA$_A$R α1 subunit expression, (3) Phasic GABAergic signaling phenotypes can be rescued by CLPs enhancing KCC2 expression and function, suggesting that impaired regulation of KCC2 by APP is responsible for these fast inhibition deficits in APP mutants, while altered tonic GABA current might involve different mechanisms (*Jo et al., 2014*; *Wu et al., 2014*). Together, these data indicate that regulation of KCC2 by APP plays a critical role in the maintenance of normal hippocampal GABAergic inhibition. Identical mIPSC amplitude in between WT and *App*$^{-/-}$ hippocampus indicates that reduction in GABA$_A$R α1 expression in *App*$^{-/-}$ does not impair basal inhibitory activity that requires activation of a portion of GABA$_A$R, which is commonly observed when correlating altered receptors with miniature activities (*Caraiscos et al., 2004*; *Chudotvorova et al., 2005*).

In previous studies, we have shown that impaired GABAergic short-term plasticity evaluated by abnormal PPR of IPSCs is due, in part, to presynaptic alterations of L-type calcium channels (LTCCs) (*Yang et al., 2009*). However, we did not see changes in the PPR of uIPSCs in the present study. This may be due to the fact that IPSCs were induced by field stimulation in previous study, which activates multiple presynaptic GABAergic neurons, and corresponding LTCC currents. By contrast, in the present study, uIPSCs were evoked by a single presynaptic action potential in a single

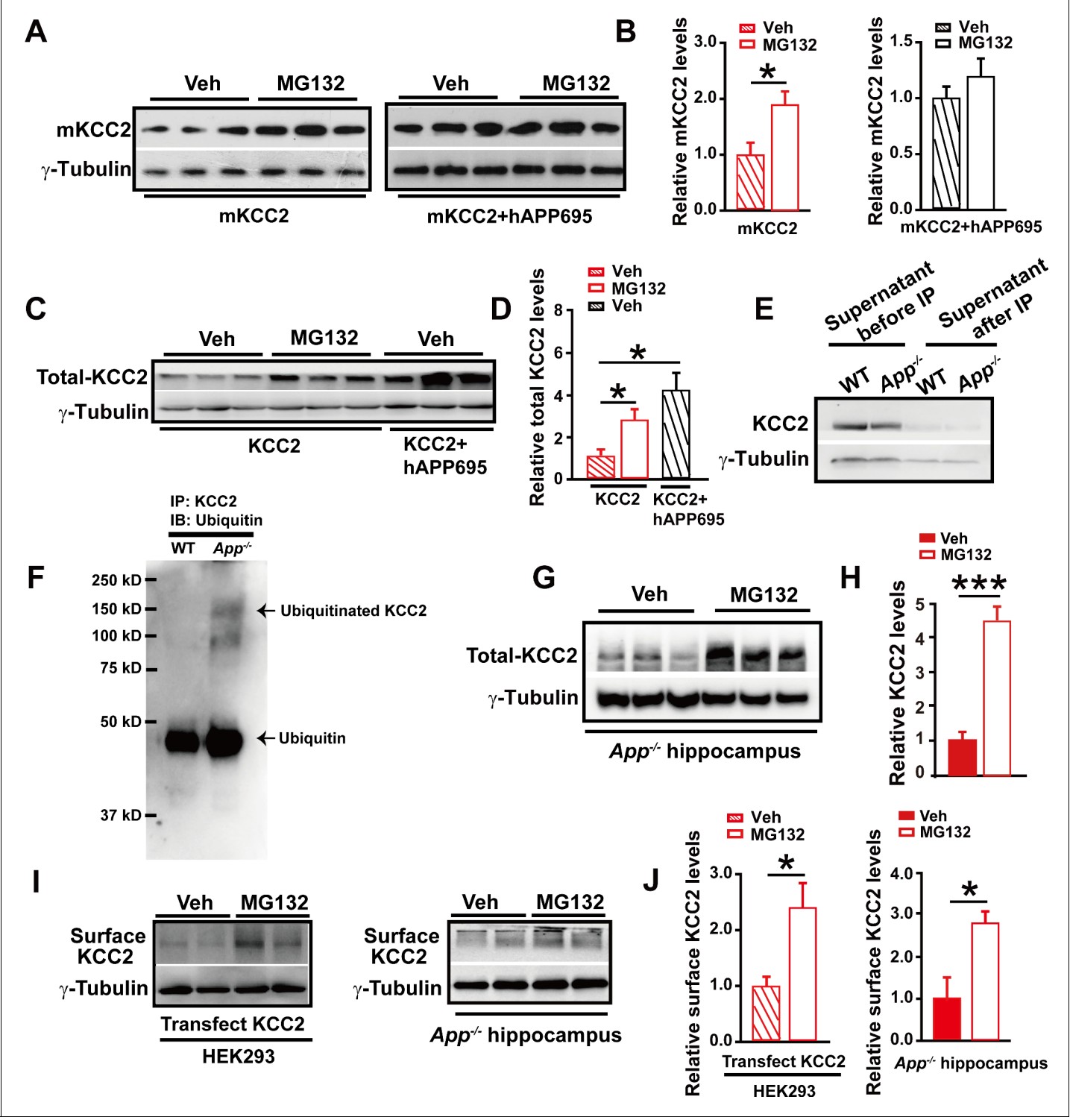

**Figure 8.** APP deficiency causes an increase in KCC2 ubiquitination leading to a reduction of KCC2 levels. (A) Representative immunoblots of HEK293 cells transfected with mKCC2 alone, mKCC2 and hAPP695, incubated with MG132 or vehicle. (B) Quantification of the immunoblots reveals a significantly increased mKCC2 levels in HEK293 cell transfected with mKCC2 alone and incubated with MG132. (C) Representative immunoblots of HEK293 cells transfected with KCC2 alone (incubated with MG132); KCC2 and hAPP695. (D) Quantification of the immunoblots reveals a significantly increased KCC2 protein levels in KCC2 alone with MG132 incubation, which is similar to co-transfection of hAPP695 and KCC2, compared to KCC2 alone incubated with vehicle. (E) Representative immunoblots of supernatant before and after IP pull down of KCC2 in WT and *App*^-/- mice. (F) Hippocampal extracts from WT and *App*^-/- littermates immunoprecipitated with an anti-KCC2 antibody (IP: KCC2) and probed with ubiquitin antibody.
*Figure 8 continued on next page*

*Figure 8 continued*

Ubiquitinated KCC2 is obviously increased in $App^{-/-}$ compared to WT mice (Each extract was from three mice per genotype, repeated twice). (G) Representative immunoblots of $App^{-/-}$ hippocampus incubated with MG132 or vehicle. (H) Quantification of the immunoblots reveals a significantly increased KCC2 protein levels in MG132 treated $App^{-/-}$ hippocampus implicating KCC2 underwent ubiquitin in the absence of APP. (I) Representative immunoblots of surface KCC2 levels of HEK293 cells and $App^{-/-}$ hippocampus incubated with MG132 or vehicle. (J) Quantification of the immunoblots reveals a significantly increased surface KCC2 levels in MG132 treated compared to vehicle control. Representative immunoblots of western blotting were from single experiment using three pairs of HEK293 cells/ or hippocampal lysates, two repeats. *p<0.05; ***p<0.001; Student's *t*-test.

The following source data is available for figure 8:

**Source data 1.** Contains source data for *Figure 8*.

GABAergic neuron such that the involvement of presynaptic LTCCs is minimized. This system allowed us to focus on postsynaptic KCC2 dysfunction in APP mutant hippocampal neurons.

KCC2 is the principle mechanism used by neurons to lower internal [Cl⁻] which is required for adequate inhibitory synaptic transmission upon activation of GABA_ARs and GlyRs (*Braat and Kooy, 2015a*; *Kaila et al., 2014*; *Rivera et al., 1999*). However, little is known about what regulates KCC2 expression and function. In an attempt to understand why KCC2 levels are diminished in APP mutants, we discovered a physical interaction between APP and KCC2. This interaction protects KCC2 from protein phosphorylation and ubiquitination which target the protein for degradation. These results suggest that APP is a novel protein binding-partner for KCC2. APP holds KCC2 on membrane through protein-protein interaction to maintain appropriate KCC2 levels, internal Cl⁻ and GABA_AR mediated inhibition. Though increased activation of glutamatergic receptors has been shown to downregulate KCC2 expression and GABA_AR mediated current (*Lee et al., 2011*), neither amplitude, frequency of mEPSCs (Amplitude; WT, 13.3 ± 0.4 pA; $App^{-/-}$, 12.6 ± 0.8 pA; p=0.5;

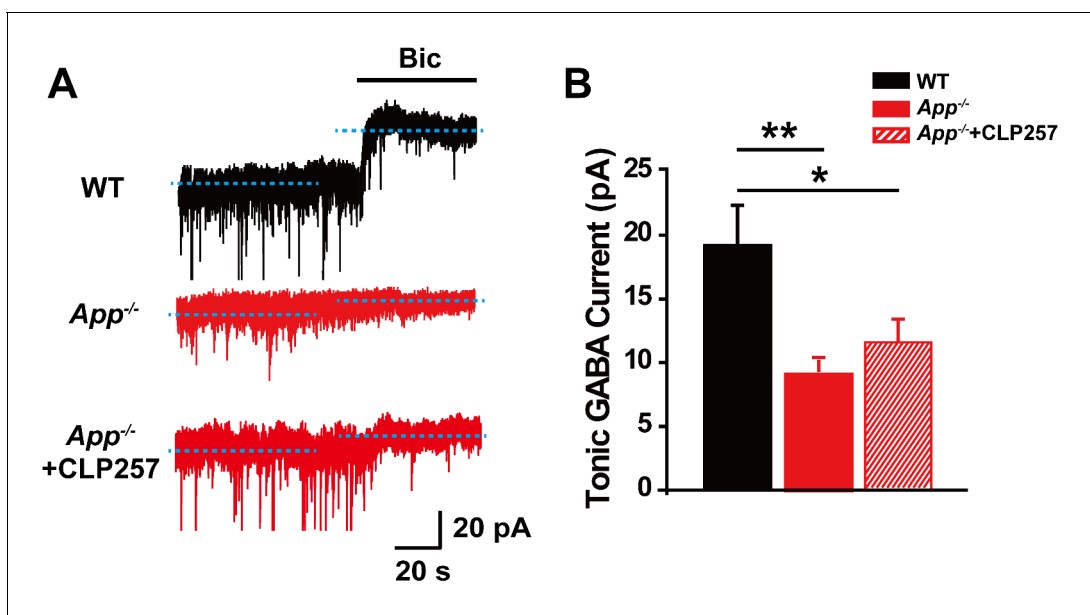

**Figure 9.** Reduced tonic GABA current in CA1 of $App^{-/-}$ hippocampus. (A) Sample traces of tonic GABA current in WT, $App^{-/-}$ and $App^{-/-}$ incubated with 100 μM CLP257. (B) Quantification of tonic GABA current amplitude shows a significant decrease in $App^{-/-}$ mice and CLP does not rescue tonic GABA current in $App^{-/-}$ hippocampus (WT, n = 10 cells from five mice; $App^{-/-}$, n = 10 cells from five mice; $App^{-/-}$+CLP257, n = 9 cells from five mice). *p<0.05; **p<0.01; one-way ANOVA.

The following source data is available for figure 9:

**Source data 1.** Contains source data for *Figure 9*.

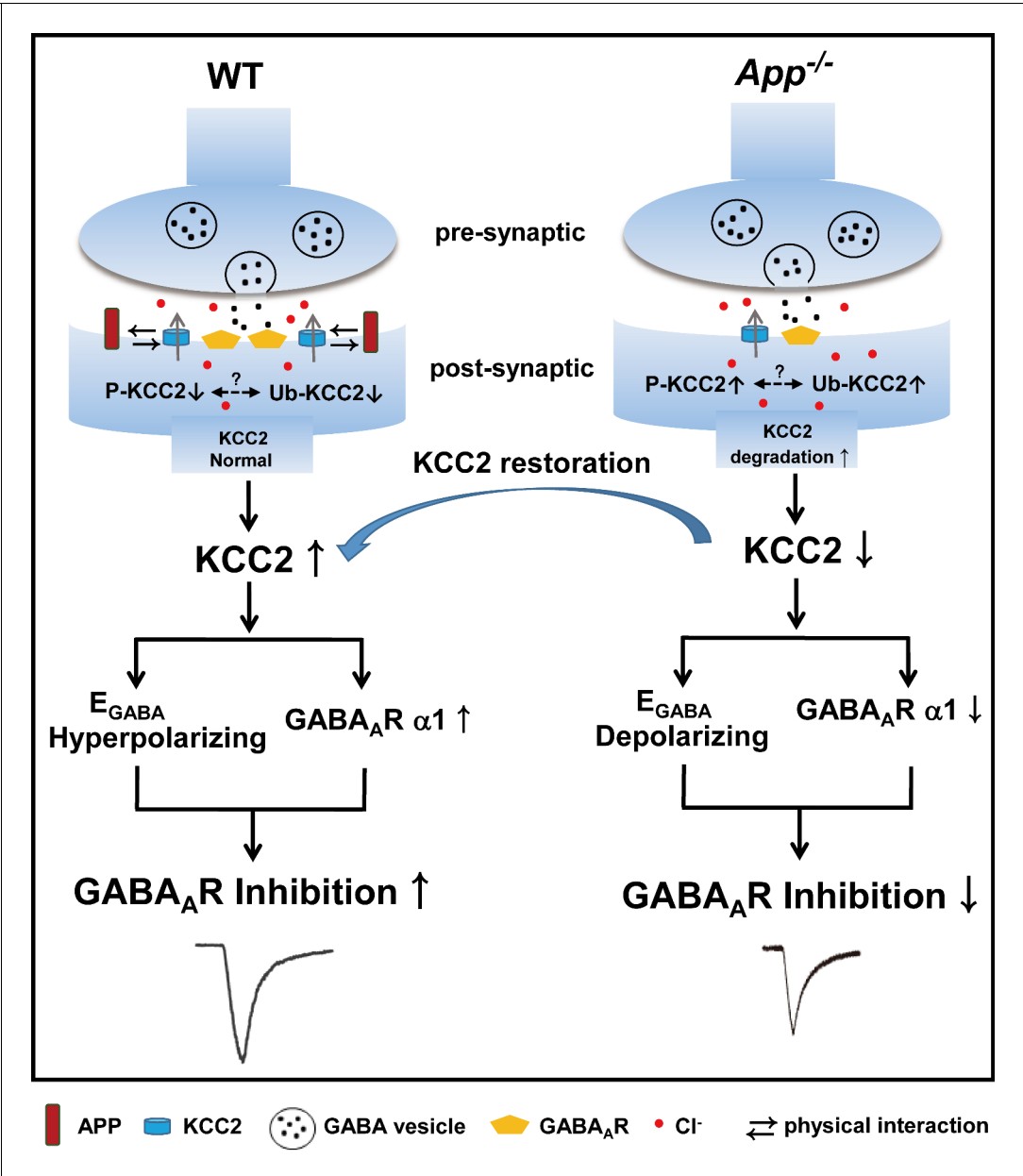

**Figure 10.** A working model shows the mechanism underlying APP regulation of GABA$_A$R mediated inhibition. KCC2 transports Cl$^-$ outside the neuron maintaining low [Cl$^-$]$_i$ in mature neurons and therefore insures an Cl$^-$ influx upon GABA binding to the GABA$_A$Rs. APP and KCC2 physically interacts to limit tyrosine phosphorylation, ubiquitination and sequential degradation of KCC2 to maintain abundant membrane KCC2 levels and hyperpolarizing E$_{GABA}$ in WT hippocampal neuron. APP deficiency leads to a loss of limit on tyrosine phosphorylation and ubiquitination of KCC2 and thus reduces KCC2 protein levels, resulting in depolarizing shift of E$_{GABA}$, reduction of GABA$_A$R α1 and GABA mediated inhibition, which can be rescued by restoration of KCC2 expression and function.

Frequency; WT, 0.2 ± 0.05 Hz; *App$^{-/-}$* 0.4 ± 0.2 Hz; p=0.3) nor GluRs protein levels changed in hippocampus of APP mutants compared to WT controls (*Figure 2—figure supplement 2A–D*, *Figure 2—source data 1*). The results suggest that excitatory synaptic activity mediated by AMPA receptors may not be affected by loss of APP at the age tested.

Changes in KCC2 levels that occur in APP mutants can have wide ranging effects on the status of GABAergic signaling. Modulation of KCC2 protein levels has been reported to affect GABA$_A$R density and GABAergic inhibition (*Chudotvorova et al., 2005*; *Succol et al., 2012*). Therefore, we

postulated that altered KCC2 function in APP mutants accounts for changes in GABAergic signaling. To test this, we pharmacologically enhanced KCC2 function in $App^{-/-}$ mice and observed a rescue in both amplitudes of isoguvacine induced currents and GABA$_A$R $\alpha$1 expression levels, indicating a rescue of GABAergic functional deficits by restoring KCC2 function. Moreover, we show that blockade of KCC2 mediated Cl$^-$ extrusion in WT hippocampus resulted in a reduction of GABA$_A$R $\alpha$1 levels as seen in $App^{-/-}$ mice and, GABA$_A$R $\alpha$1 levels are positively correlated with KCC2 expression and function during early stage of development. We thus reasoned that increase in KCC2 mediated Cl$^-$ extrusion might be responsible for upregulation of surface GABA$_A$R $\alpha$1 levels.

Though the exact mechanism underlying how restoring KCC2 rescued surface GABA$_A$R $\alpha$1 levels remains unclear, given that APP interacts directly with KCC2, and increasing KCC2 expression and function rescued GABAergic signaling phenotypes in $App^{-/-}$ hippocampus, we propose that KCC2 is the primary defect caused by APP deletion to alter GABAergic signaling, while changes in GABA$_A$R $\alpha$1 levels and GABA$_A$R mediated inhibition are secondary to altered KCC2 expression and function.

Our results indicate that full length APP is required to maintain normal levels of KCC2 protein. Together with a recent study of the $App\Delta$CT15-DM mice, a mouse model lacking the last 15 amino acids of APP as well as the APP homologue, APLP2, which demonstrated impairments of synaptic plasticity and hippocampal-dependent behavior (*Klevanski et al., 2015*), it is likely that APP proteolysis may be a mechanism for turning off normal APP functions which needs to be taken into account when exploring the pathophysiology of APP with respect to KCC2 regulation.

GABAergic signaling has been shown to be disrupted in many diseases of the nervous system (*Braat and Kooy, 2015a*), including fragile X syndrome and autism spectrum disorder (*Braat and Kooy, 2015b*; *Han et al., 2014*). Changes in KCC2 function may account for many GABAergic signaling deficits, as it has been shown to be involved in the pathogenesis of epilepsy, schizophrenia, autism and aging brain (*Ben-Ari et al., 2012*; *Ferando et al., 2016*; *Tang et al., 2016*; *Tao et al., 2012*). Though the GABAergic system appears to be resistant to A$\beta$ toxicity (*Selkoe, 2002*), recent research has demonstrated dysfunctional GABAergic inhibition in AD (*Verret et al., 2012*). In the present study, we have described a cellular mechanism by which loss of APP function downregulates KCC2 to impair GABAergic signaling. Given the results presented here in APP mutant hippocampal neurons, if APP function is compromised in the context of progressing AD, this may lead to the gradual loss of GABAergic signaling capacity in the hippocampus and associated changes in memory function. Together with a most recent study revealing a critical role of KCC2 in information storage in aging brain (*Ferando et al., 2016*), the present study suggests that KCC2 might serve as a therapeutic target to improve GABAergic inhibition to potentially restore hippocampal function and improve memory in AD.

## Materials and methods

### Animals and plasmids

All experiments and animal housing were in accordance with procedures approved by the Ethics Committee for animal research at South China Normal University, according to the Guidelines for Animal Care established by the National Institute of Health. Both female and male mice were used in this study. $App^{-/-}$ and wild type (WT) mice (*Zheng et al., 1995*) were generated by crossing $App^{+/-}$ males and females obtained from Model Animal Research Center of Nanjing University (Nanjing, China). $App^{-/-}$ mice were crossed with $Gad67^{+/GFP}$ mice (*Tamamaki et al., 2003*) (gift from Dr. Yuqiang Ding at Tongji University School of Medicine, Shanghai, China) when necessary to generate $App^{-/-}$-$Gad67^{+/GFP}$ and $WT$-$Gad67^{+/GFP}$ controls. Soluble APP ectodomain beta (sApp$\beta$) knock in mice (*Li et al., 2010*), full length human APP (hAPP695), APP C-terminal fragments (C99) and sAPP$\beta$ constructs were from Hui Zheng lab at Baylor College of Medicine. The KCC2 construct was gift from John A Payne (School of Medicine, University of California). The GABA$_A$R $\alpha$1 construct was gift from Gangyi Wu (South China Normal University, Guangzhou, China). CYLD construct was gift from Shaocong Sun (University of Texas MD Anderson Cancer Center, Texas, USA). GABA$_A$R $\beta$2 construct was from Yong Zhang (Peking University).

## Neuronal culture

Cultured hippocampal neurons were prepared as described (*Yang et al., 2009*) with some modifications. Briefly, we dissected hippocampus from *App$^{-/-}$-Gad67$^{+/GFP}$* and *WT-Gad67$^{+/GFP}$* littermates of P0/1 pups, trypsinized the tissue at 37°C for 30 min, and then gently triturated in culture medium containing 10% heat-inactivated fetal bovine serum using fire-polished pipettes, before centrifuging at 1000 rpm for 10 min. Cells were resuspended in Neurobasal Medium supplemented with B27 and L-glutamine (Invitrogen, Carlsbad, CA), and plated at 1–2 × 10$^4$ cells/ml onto 13 mm poly-L-lysine coated coverslips. One day after plating, 4 M cytosine-D-arabinofuranoside (Ara-C) (Sigma) was added to prevent astrocyte proliferation. Cultures of 12–14 DIV were used for patch clamp recordings.

## Electrophysiology

### Whole-cell recording in acute hippocampal slices

Hippocampal slices (350 μm thick) were cut using a vibratome (LeicaVT1000S) in ice-cold cutting solution containing the following: 25 mM NaHCO$_3$, 2.5 mM KCl, 7 mM MgCl$_2$, 1.25 mM NaH$_2$PO$_4$, 7 mM Glucose, 0.5 mM CaCl$_2$, 115 mM Choline Cl. Slices were incubated for 45 min in artificial cerebrospinal fluid (aCSF) solution containing the following: 119 mM NaCl, 2.5 mM KCl, 2.5 mM CaCl$_2$, 1.3 mM MgSO$_4$, 1 mM NaH$_2$PO$_4$, 11 mM D-glucose, 26.2 mM NaHCO$_3$. Pipettes were filled with an internal solution of 136.5 mM K-Gluconic acid, 17.5 mM KCl, 9 mM NaCl, 1 mM MgCl$_2$, 0.2 mM EGTA, 10 mM HEPES (pH 7.3) containing gramicidin A 10 μg/ml (Sigma; diluted from a stock solution of 10 mg/ml in DMSO). The recording aCSF contains: 140 mM NaCl, 4.7 mM KCl, 2.5 mM CaCl$_2$, 1.2 mM MgCl$_2$, 11 mM D-glucose, 10 mM HEPES (pH 7.3). Both incubating and recording solutions were bubbled with 95% O$_2$ and 5% CO$_2$ throughout the experiment. We performed all the experiments for the determination of E$_{GABA}$ values in the presence of TTX (1 μM), CNQX (20 μM), APV (50 μM) and Strychnine (1 μM) to block voltage-gated sodium channels, AMPA receptors, NMDA receptors, and GlyRs, respectively. A Picospritzer (Parker Instrumentation) was used to eject GABA (100 μM, 100 ms) directly onto neuronal soma through a fine pipette (~2 μm tip). For mEPSC and mIPSC recordings, the aCSF was supplemented with 1 μM TTX, and pipettes were filled with an internal solution of 100 mM Cs gluconate, 5 mM CsCl, 10 mM HEPES, 2 mM MgCl$_2$, 1 mM CaCl$_2$, 11 mM BAPTA, 4 mM ATP, 0.4 mM GTP. To record tonic GABA current, 10 μM CNQX and 50 μM APV were added into the aCSF to block glutamate receptors. The total tonic GABA current was revealed by the change of holding current after perfusion of 100 μM bicucullin (Bic). Pipettes were filled with an internal solution of 135 mM CsCl, 5 mM Na-phosphocreatine, 10 mM HEPES, 5 mM EGTA, 4 mM MgATP, 5 mM QX-314 and 0.5 mM Na$_2$GTP (pH 7.3) (*Wu et al., 2014*). Data were filtered during acquisition with a low pass filter set at 2 kHz using pClamp10 (Molecular Devices). Data analysis was performed offline with Clampfit 10.2 (Molecular Devices).

### Whole-cell recording in hippocampal cultures

GABA reversal potential recordings were made using a perforated patch-clamp technique. Pipettes were filled with an internal solution of 140 mM KCl and 10 mM HEPES (pH 7.3) containing gramicidin A 50 μg/ml (Sigma; diluted from a stock solution of 50 mg/ml in DMSO). Data were filtered during acquisition with a low pass filter set at 2 kHz using pClamp10 (Molecular Devices). Data analysis was performed offline with Clampfit 10.2 (Molecular Devices).

### Dual whole-cell patch-clamp recordings

The culture coverslips were transferred into a heated (30°C) submerged recording chamber, continuously perfused (2–3 ml/min) with tyrode buffer: 129 mM NaCl, 5 mM KCl, 0.01 mM Glycine, 1 mM MgCl$_2$, 2 mM CaCl$_2$, 25 mM HEPES, 30 mM D-Glucose. We performed all experiments for the determination of unitary inhibitory postsynaptic current (uIPSC), and paired-pulse ratio (PPR, uIPSC2/uIPSC1) values in the presence of CNQX (20 μM) and APV (50 μM) to block AMPA receptors and NMDA receptors, respectively. GABAergic neurons were patched in whole-cell, current-clamp configuration using an Axopatch 700B amplifier (Axon Instruments) Channel 1. The pipette solution contained 110 mM K-Gluconic acid, 10 mM NaCl, 1 mM MgCl$_2$, 10 mM EGTA, 40 mM HEPES, 2 mM ATP, 0.3 mM GTP (pH adjusted to 7.2 with NaOH). Pyramidal neurons were patched in voltage-clamp configuration and recorded at a holding potential of −60 mV using an Axopatch 700B

amplifier (Axon Instruments) Channel 2. The pipette solution contained 100 mM Cesium Methanesulfonate, 10 mM NaCl, 10 mM TEA-Cl, 1 mM $MgCl_2$, 10 mM EGTA, 40 mM HEPES, 2 mM ATP, 0.3 mM GTP, 4 mM QX-314 (pH adjusted to 7.2 with NaOH). Before uIPSC recordings, voltage responses of presynaptic and postsynaptic cells were recorded by applying long hyperpolarizing and depolarizing current pulse (300 ms) injections to examine basic electrophysiological properties, including input resistance, single spike kinetics, voltage-current relationships, repetitive firing patterns, and firing frequency. For evoked GABA current recording, a Picospritzer (Parker Instrumentation) was used to eject isoguvacine (100 μM, 100 ms) directly onto neuronal soma through a fine pipette (~2 μm tip).

## Quantitative Real-time PCR (qRT-PCR)

Mice brains were dissected and immediately frozen in liquid nitrogen. Total RNA was extracted using Trizol reagent. RNA pellets were resuspended in diethylpyrocarbonate-treated water and RNA concentration was measured using a Nanodrop2000c spectrometer (Thermo Scientific). RNA was DNase treated (DNase I, amplification grade, Invitrogen) and then reverse-transcribed using the Superscript III First-Strand Synthesis System (Invitrogen). Relative quantification of gene expression was performed using Taq Manprobes and an ABI Prism 7000 Sequence Detection System. Platinum Quantitative PCR Super Mix-UDG w/ROX (Invitrogen) was used with the following primers and probes: *App* F-primer, (5′–3′) CCAAGAGGTCTACCCTGAACTGC; *App* R-primer, (5′–3′) AGG-CAACGGTAAGGAATCACG; *Actβ* F-primer (5′–3′) GTGACGTTGACATCCGTAAAGA; *Actβ* R-primer (5′–3′) GTAACAGTCCGCCTAGAAGCAC; *Slc12a5* F-primer (5′–3′) GGGCAGAGAGTACGATGGC; *Slc12a5* R-primer (5′–3′) TGGGGTAGGTTGGTGTAGTTG. Assay efficiencies were experimentally determined using a five-point dilution series of cDNA spanning a 100-fold range in concentration. 0.025 μg cDNA template was used per reaction. Statistical analysis was performed on $2^{-\Delta\Delta Ct}$ values.

## Immunohistochemistry

Mice were anesthetized with intraperitoneal injection of 20% urethane (0.01 ml/g) and perfused transcardially with 0.9% physiological saline and 4% paraformaldehyde (PFA) in phosphate buffer (0.01 M PBS, pH 7.4). Following perfusion, the brains were dissected and post-fixed overnight in 4% PFA followed by a sucrose series (10%, 20% and 30%) solution at 4°C for cryoprotection. Serial coronal/sagittal sections of tissue were obtained using Lycra frozen slicer 1800 with a thickness of 30 μm. Free-floating sections were rinsed in PBS three times for 5 min and processed for antigen retrieval by boiling in 10 mM citrate buffer (pH 6.0) for 5 min. The sections were incubated in 3% $H_2O_2$ in PBS for 15 min at room temperature (RT) to quench endogenous peroxidases, washed three times for 5 min and subsequently incubated in 0.3% Triton X-100 in PBS for 2 hr. To block nonspecific binding, sections were incubated in 3% bovine serum albumin (BSA) for 30 min. Sections were then incubated overnight with primary antibody against KCC2 (Neuromab, 75–013; Millipore, 07–432) at 4°C. After three washes with PBS, secondary antibody (CWS) application was performed at RT for 2 hr followed by three additional washes with PBS for 10 min. The secondary antigen was visualized with 3,3-diaminobenzidine tetrahydrochloride (DAB, Sola). The reaction was terminated by rinsing in PBS and then sections were mounted on gelatin-subbed slides, dehydrated in ascending alcohol concentrations, cleared through xylenes, and covered with DPX resin. KCC2 immunofluorescent staining was conducted as described (*Yang et al., 2009*).

## Western blotting

Mouse brains were homogenized using SDS lysis buffer (50 mM Tris pH 7.5, 150 mM NaCl, 5 mM EDTA pH 8.0, 1% SDS) containing complete protease inhibitor cocktail (Roche). After 1 min of homogenization, the cellular debris was removed by centrifugation at 14000 rpm for 10 min at 4°C and supernatant was collected and denatured for 20 min at 75°C. Tissue lysates were subjected to SDS-PAGE (Bio-Rad) and transferred to nitrocellulose membranes. Membranes were blocked for 0.5 hr using 5% non-fat dry milk in TBS containing 0.5% Tween-20 (TBST), then incubated with specific primary antibodies against KCC2 (Neuromab, 75–013; Millipore, 07–432; Santa Cruz, sc-19419), NKCC1 (Abcam, ab191289), APP (Abcam, ab15272), 4G10 (Millipore, 16–452), PLIC-1 (ABGENT, AP2176c), GABA$_A$R subunits: α1 (Millipore, 06–868), α5 (Abcam, ab175195), β2 (Abcam, ab156000), δ (Abcam, ab110014), γ2 (Abcam, ab87328), GluR1 subunit (Abcam, ab31232) and GluR2 subunit

(Abcam, ab206293) of AMPA receptors, and ubiquitin (Abcam, ab7254). Anti γ-tubulin antibody (Sigma, T6557) was used as a loading control. Antibodies were diluted at 1:1000 to 1:2000 at use. After three washes with TBST, secondary antibody (CWS) application was performed at RT for 1 hr using 5% milk in TBST followed by three additional washes with TBST. Bands were visualized using Immobilon Western ECL system and analyzed with Gel Pro Analysis software. For surface KCC2 detection, biotinylation of cell surface proteins was performed as described previously (*Lee et al., 2010*). Briefly, HEK293 cells transfected with KCC2 constructs or cultured neurons were washed two times with PBS containing 0.5 mM $MgCl_2$ and 1 mM $CaCl_2$ (PBS-CM) and then incubated with 2 ml of PBS-CM containing 1 mg/ml Sulfo-NHS-SS-Biotin (Pierce) for 30 min at 4°C. After labeling, the biotin reaction was quenched by washing three times with ice-cold PBS-CM containing 50 mM glycine and 0.1% BSA. Cells were then lyzed with 1 ml lysis buffer and the lysate was collected. Protein concentration was determined using Micro BCA protein assay kit (Thermo Scientific). After correction for protein concentration, 40 µl of 1:1 slurry of UltraLink NeutrAvidin (Pierce) was added to the lysates to pull down the biotin-labeled surface proteins for 2 hr at 4°C. The NeutrAvidin beads were washed and bound materials were analyzed by SDS-PAGE.

## Molecular cloning

To obtain a KCC2 plasmid in which the tyrosine phosphorylating sites, Y903A and Y1087A, were mutated (mKCC2), the KCC2 open reading frame (ORF) was amplified from the pCAGIG vector, incorporate Y903A and Y1087A mutation using the stratagene QuickChange method as described (*Lee et al., 2010*). For mutation of Y903 the following primers were used: (sense) TCCGGATCCAC TCGAGATGCTCAAC; (antisense) AATGTCTTCTCGAAGGTG TATGCTG. For mutation of Y1087 the following primers were used: (sense) TACACCTTCGAGAAGACATTGG; (antisense): ATTTACGTAG CGGCCGCTCAGGAGT AGATGGATATCTGCAGAATTCCATGAAGTTTTCATCTCC.

## Transfection

Commercially-available HEK293 cells, original transformated as reported (*Graham et al., 1977*), were co-transfected with KCC2 and GABA$_A$R α1 or β2; KCC2 and APP/ or different APP fragments. Whenever necessary, KCC2 was replaced by mKCC2. Transfected HEK293 cells were washed with ice-cold PBS, centrifuged at 5000 rpm for 10 min at 4°C, and the precipitate was collected and homogenized using SDS lysis buffer, centrifuged at 14,000 rpm for 10 min at 4°C and supernatant and denatured for 20 min at 75°C. Supernatants were analyzed by western blotting.

## Co-immunoprecipitation (Co-IP)

HEK293 cells were grown in DMEM (high glucose with L-glutamine) containing 10% FBS. For immunoprecipitation, HEK293 cells were plated onto 10 cm culture dishes, and grown to 70% confluency. Cell transfections were performed with lipofectamine 2000 (Invitrogen), following the manufacturer's recommended protocol. 24–48 hr after transfection, cells were washed once with cold PBS then homogenized with 1 ml cold lysis buffer (50 mM Tris pH 7.5, 150 mM NaCl, 1 mM EDTA, 1% NP40, and protease inhibitor) in 2 ml tubes. To remove cell debris, lysates were centrifuged for 20 min at 14,000 rpm at 4°C. Supernatants were incubated with 2 µl of KCC2 (Millipore, 07–432), APPc (*Wang et al., 2007*), or GABA$_A$R α1 (Millipore, 06–868), antibodies at 4°C overnight. Protein A agarose beads at 30 µl (Millipore) was added to each sample and they were incubated at 4°C for 2–3 hr. To control for nonspecific binding, protein lysates incubated with rabbit or mouse IgG and beads were processed in parallel. Subsequently, beads were washed three times with 500 µl of cold lysis buffer. Immunoprecipitates were eluted from the beads by adding 30 µl of SDS-PAGE sample buffer and heating for 20 min at 75°C. The eluates (15 µl) were analyzed by western blotting. For hippocampal lysates immunoprecipitation, we followed the HEK293 cells protocol above. Coomassie blue staining was conducted as described (*Savas et al., 2015*).

## Proximity ligation assay (PLA)

HEK293 cells were grown in DMEM containing 10% FBS. For PLA (*Söderberg et al., 2006*), HEK293 cells were plated onto 10 cm culture dishes, and grown to 70% confluency. On the day of experiment, the wells were washed with 1xPBS and then added with 4% PFA and incubated for 10 min at RT without agitation. Then the cells were washed with PBS for 3 times, 5 min each, with agitation

and treated with 0.5% Triton X-100 in PBS (PBST) for 10 min without agitation. Wash the cells with TBST for 3 times, 5 min each, with agitation. Block with Duolink II Blocking Solution (1x) for 1 hr at 37°C and then incubate with specific primary antibodies against KCC2 (Neuromab, 75–013) or APP (Abcam, ab15272). After wash with 1x Duolink II Wash Buffer A, cells were incubated with two PLA probes (Duolink II anti-Mouse MINUS and Duolink II anti-Rabbit PLUS) 1:5 in Antibody Diluent. Wash the slides in 1x Wash Buffer A for 2 times, 5 min each, with agitation at RT. Incubate the slides with Ligation-Ligase solution for 30 min at 37°C. Wash the slides in 1x Wash Buffer A with for 2 times, 2 min each. Then incubate the slides with Amplification-Polymerase solution for 100 min at 37°C. Wash the slides for 2 times, 10 min each, in 1x Duolink II Wash Buffer B and incubate with DAPI for 15 min at 37°C. Data was analyzed with a fluorescence microscope.

## KCC2 restoration experiment

KCC2 enhancers, CLP257 and CLP290 (*Gagnon et al., 2013*) were gifts from Yves De Koninck (Institut universitaire en santé mentale de Québec, Qc, Canada).

CLP257 incubation: CLP257 was dissolved in 200 µl DMSO as 100 mM stock solution and diluted using aCSF to 100 µM. For slice recording, hippocampal slices (350 µm thick) of $App^{-/-}$ mice were cut using a vibratome (LeicaVT1000S) as described above for whole-cell recording in acute hippocampal slices. Slices were incubated for 2 hr in aCSF containing 100 µM CLP257 or DMSO and babbled with 95% $O_2$ and 5% $CO_2$. For surface and total KCC2 protein levels evaluation, HEK293 cells, which was transfected with KCC2 were incubated for 2 hr in 100 µM CLP257 or DMSO.

CLP290 intraperitoneal injection: CLP290 was dissolved in HPCD and intraperitoneally administered once daily at a dose of 100 mg/kg for one week. HPCD injection group was used as vehicle control.

## Statistical analysis

Unless otherwise stated, data are displayed as mean±SEM. Student's *t*-test was used for statistical analysis in two-group comparison. For comparison among multiple groups, one-way ANOVA or two-way ANOVA with post hoc tests were used. Statistical significance was set at $p < 0.05$.

## Acknowledgements

The authors thank Dr. Yasunori Hayashi for suggestions on the manuscript; Zhijie Zhang for technical support. This work was supported by grants from the National Natural Science Foundation of China (31171018, 31171355), the Science and Technology Division of Guangdong (2013KJCX0054, 201607010320), and the Natural Science Foundation of Guangdong Province (2014A030313418, 2014 A030313440).

## Additional information

### Funding

| Funder | Grant reference number | Author |
| --- | --- | --- |
| National Natural Science Foundation of China | 31171018 | Cheng Long<br>Li Yang |
| Natural Science Foundation of Guangdong Province | 2014A030313418 | Cheng Long<br>Li Yang |
| National Natural Science Foundation of China | 31171355 | Cheng Long<br>Li Yang |
| Natural Science Foundation of Guangdong Province | 2014A030313440 | Cheng Long<br>Li Yang |
| The Science and Technology Division of Guangdong Province | 2013KJCX0054 | Li Yang |
| The Science and Technology Division of Guangdong Province | 201607010320 | Li Yang |

The funders had no role in study design, data collection and interpretation, or the decision to submit the work for publication.

## Author contributions

MC, JW, JJ, XZ, KW, YL, QH, JZ, HL, NL, YW, Acquisition of data, Analysis and interpretation of data; NJJ, Acquisition of data, Drafting or revising the article; XR, Worked on HEK293 cell culture and transfection; HZ, CL, Analysis and interpretation of data, Contributed unpublished essential data or reagents; LY, Conception and design, Drafting or revising the article

## Author ORCIDs

Li Yang, http://orcid.org/0000-0001-7448-8588

## Ethics

Animal experimentation: This study was conducted along the guidelines described in "The Guide for the Care and Use of Laboratory Animals" (Eighth Edition). All of the animals were handled according to approved institutional animal care and use committee (IACUC) protocols (#0312-11) of the South China Normal University. Every effort was made to minimize suffering.

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
