## [Decision Letter]

Thank you for submitting your article "APP is a novel protein-binding partner for KCC2: An insight on the role of APP in hippocampal GABAAR mediated inhibition" for consideration by *eLife*. Your article has been favorably evaluated by a Senior Editor and three reviewers, one of whom is a member of our Board of Reviewing Editors. The following individual involved in review of your submission has agreed to reveal his identity: C Justin Lee (Reviewer #2).

The reviewers have discussed the reviews with one another and the Reviewing Editor has drafted this decision to help you prepare a revised submission.

Summary:

This study provides some evidence that APP stabilizes KCC2, a K^+^-Cl^-^ cotransporter, on the cell surface and that APP regulates GABAA receptor-mediated synaptic transmission. The authors first showed the GABA reversal potential shifts toward depolarization, the decreased unitary IPSC amplitude, and the reduced GABAA receptor protein level in the hippocampus of APP-/- mice. Then, the authors found that APP directly binds to KCC2 and prevents KCC2 from its degradation. Furthermore, the authors showed that restoration of KCC2 function in APP-/- mice by CLP290 rescues GABAA receptor levels and function. Because patho-physiological functions of APP have attracted considerable attention, this work addresses a generally important topic. Addressing the following points would strengthen this paper.

Essential revisions:

1) The authors provide data suggesting that KCC2 in the absence of APP is degraded through ubiquitin-dependent mechanisms and enhanced endocytosis. However, the data supporting this conclusion mainly presented in Figure 6 are very weak. Specifically, there is no evidence supporting that APP2 deficiency causes KCC2 ubiquitination or KCC2 endocytosis. In addition, it is unclear whether endocytosis of KCC2 is functionally coupled to ubiquitination.

2) Although the link between APP and KCC2 is abundantly clear, the link between KCC2 and GABAA is not so clear. For example, in Figure 1 and Figure 1—figure supplement 1 there appears to be a significant difference in Egaba between WT and APP-/-. However, there appears to be a difference in the conductance (slope of the IV) between whole cell (Figure 1) in slice and perforated patch (Figure 1—figure supplement 1) in culture. This type of discrepancy is also shown in Figure 2—figure supplement 1, in which the amplitude of mIPSC for WT and APP-/- did not show the expected difference. These discrepancies might be due to the fact that authors apparently do not fully address the field of GABA inhibition. GABA has not only phasic mode but also tonic mode of action. In recent studies, tonic GABA release from reactive astrocytes and its inhibitory action has been implicated in Alzheimer's disease (Jo, et al., 2012; Wang, et al., 2012; also, see Yarishkin et al., 2015). KCC2 could not only affect synaptic GABAA receptors but also extrasynaptic GABAA receptors. In addition, isoguvacine which is used for monitoring functional surface GABAA activates not only synaptic GABAA but also extrasynaptic GABAA, complicating authors' interpretations. Therefore, authors need to introduce the current state of phasic and tonic GABA appropriately and discuss their results accordingly.

---

## [Author Response]

*Essential revisions:*

*1) The authors provide data suggesting that KCC2 in the absence of APP is degraded through ubiquitin-dependent mechanisms and enhanced endocytosis. However, the data supporting this conclusion mainly presented in Figure 6 are very weak. Specifically, there is no evidence supporting that APP2 deficiency causes KCC2 ubiquitination or KCC2 endocytosis. In addition, it is unclear whether endocytosis of KCC2 is functionally coupled to ubiquitination.*

We would like to thank the reviewers for these questions about the KCC2 ubiquitination phenotype in APP^-/-^ mice. In our original submission, we showed that, in HEK293 cells, mKCC2 (a mutant form of KCC2 in which tyrosine phosphorylation is inhibited) underwent ubiquitination and sequential degradation in the absence of APP co-transfection based on experiment showing that mKCC2 levels could be rescued by co-transfection with CYLD (a deubiquitinating enzyme) construct (previously Figure 6, now Figure 7), implicating ubiquitination as a mechanism potentially leading to KCC2 degradation. In our new experiments, we have confirmed, by treating transfected HEK293 cells (mKCC2 alone, mKCC2+APP; KCC2 alone, KCC2+APP) with the ubiquitination inhibitor MG132 (or vehicle), that blockade of ubiquitination by MG132 significantly increased both KCC2 and mKCC2 levels in the absence of APP (Figure 8 in the revised manuscript).

These data in HEK293 cells implicate that APP deficiency might result in enhanced KCC2 ubiquitination, which contributes, at least in part, to reduced KCC2 levels in APP^-/-^ hippocampus. To determine whether KCC2 ubiquitination occurs in brain lysates as well, we immunoprecipitated KCC2 from APP^-/-^ and WT mouse hippocampal lysates and immunoblotted for ubiquitin/ubiquitinated protein to evaluate differences in KCC2 ubiquitination in APP^-/-^ and WT. In APP^-/-^ extracts, we observed significantly increased levels of ubiquitinated KCC2, indicating increased KCC2 ubiquitination in APP^-/-^ hippocampus (Figure 8). Furthermore, KCC2 levels were significantly increased in MG132-treated APP^-/-^ hippocampus compared to DMSO controls (Figure 8). Moreover, MG132 treatment increased KCC2 surface levels in both HEK293 cells and APP^-/-^ hippocampus (Figure 8), implicating that KCC2 endocytosis might be functionally coupled to ubiquitination. Together, these results support the hypothesis that APP deficiency does indeed cause KCC2 ubiquitination. Please see subsection “APP also functions to suppress ubiquitination and thus degradation of KCC2” and the Discussion, second paragraph.

*2) Although the link between APP and KCC2 is abundantly clear, the link between KCC2 and GABAA is not so clear. For example, in Figure 1 and Figure 1—figure supplement 1 there appears to be a significant difference in Egaba between WT and APP-/-. However, there appears to be a difference in the conductance (slope of the IV) between whole cell (Figure 1) in slice and perforated patch (Figure 1—figure supplement 1) in culture. This type of discrepancy is also shown in Figure 2—figure supplement 1, in which the amplitude of mIPSC for WT and APP-/- did not show the expected difference. These discrepancies might be due to the fact that authors apparently do not fully address the field of GABA inhibition. GABA has not only phasic mode but also tonic mode of action. In recent studies, tonic GABA release from reactive astrocytes and its inhibitory action has been implicated in Alzheimer's disease (Jo, et al., 2012; Wang, et al., 2012; also, see Yarishkin et al., 2015). KCC2 could not only affect synaptic GABAA receptors but also extrasynaptic GABAA receptors. In addition, isoguvacine which is used for monitoring functional surface GABAA activates not only synaptic GABAA but also extrasynaptic GABAA, complicating authors' interpretations. Therefore, authors need to introduce the current state of phasic and tonic GABA appropriately and discuss their results accordingly.*

We greatly appreciate the advice that we do a more extensive phenotypic analysis to introduce the current state of phasic and tonic GABA appropriately and we feel that this has greatly improved the manuscript. We have previously shown that APP deficiency results in an impairment in tonic GABA current in dentate gyrus (Wang et al., 2014), and we have now added a figure showing significantly reduced tonic GABA current, which was evaluated by application of the GABA_A_R antagonist bicuculline, in CA1 of APP^-/-^ mice. Interestingly, restoring KCC2 activity with CLP257 did not rescue the impaired tonic GABA current in APP^-/-^ hippocampus (Figure 9). In addition, using western blotting, we have now shown that there are no significant differences between WT and APP^-/-^ mice in levels of GABA_A_R α5 and δ subunits, which mediate tonic GABA current (Brickley & Mody 2012; Caraiscos et al., 2004; Curia et al., 2009) (data added to Figure 2 in the revised manuscript). These results suggest that altered tonic GABA current in APP-deficient CA1 may involve presynaptic mechanisms, e.g., abnormal astrocytic GABA activities, that require further investigation in the future. The tonic GABA data have been added to the Results and Discussion, respectively.

We apologize for our lack of clarity. Actually, both Figure 1 (in slice) and Figure 1—figure supplement 1 (in culture) are perforated patch experiments. In addition, the differences in conductance between brain slice and culture seen in our present study might also be explained, in part, by E_GABA_ differences between brain and culture (Khirug et al., 2005), different internal/external solutions and different perfusion speeds used for recording E_GABA_ in slice and culture.

Please see the subsection “uIPSC amplitude and GABAAR α1 subunit levels are reduced in App-/- hippocampus” of the manuscript for an explanation of the identical amplitude of mIPSC in WT and APP^-/-^.